# Uncertainty-Aware Transformers: Conformal Prediction for Language Models

## Abstract

Transformers have had a profound impact on the field of artificial intelligence, especially on large language models and their variants. Unfortunately, as was the case historically with neural networks, the black-box nature of transformer architectures presents significant challenges to interpretability and trustworthiness. These challenges generally emerge in high-stakes domains, such as healthcare, robotics, and finance, where incorrect predictions can have significant negative consequences, such as misdiagnosis or failed investments. For models to be genuinely useful and trustworthy in critical applications, they must provide more than just predictions: They must supply users with a clear understanding of the reasoning that underpins their decisions. This article presents an uncertainty quantification framework for transformer-based language models. This framework, called CONFIDE (CONformal prediction for FIne-tuned DEep language models), applies conformal prediction to the internal embeddings of encoder-only architectures, like BERT and RoBERTa, based on hyperparameters, such as distance metrics and principal component analysis. CONFIDE uses either [CLS] token embeddings or flattened hidden states to construct class-conditional nonconformity scores, enabling statistically valid prediction sets with instance-level explanations. Empirically, CONFIDE improves test accuracy by up to 4.09% on BERT-tiny and achieves greater correct efficiency (i.e., the expected size of the prediction set conditioned on it containing the true label) compared to prior methods, including NM2 and VanillaNN. We show that early and intermediate transformer layers often yield better-calibrated and more semantically meaningful representations for conformal prediction. In resource-constrained models and high-stakes tasks with ambiguous labels, CONFIDE offers robustness and interpretability where softmax-based uncertainty fails. When the exchangeability condition is violated, no method we tested, including CONFIDE, achieves nominal coverage; minority or ambiguous classes often have undercoverage. We, therefore, position CONFIDE as a framework for practical diagnostic and efficiency/robustness improvement over prior conformal baselines.

## 1 Introduction

Transformer architectures, first introduced in Vaswani et al. (2017), revolutionized natural language processing (NLP) by introducing the self-attention mechanism, enabling models to capture complex contextual relationships. The introduction of BERT (Bidirectional Encoder Representations from Transformers) further enabled transformer progress by showcasing the effectiveness of encoder-only architectures across diverse linguistic tasks, from sentiment analysis to named entity recognition (Devlin et al., 2019).

A range of encoder-based transformer variants followed, each optimizing different aspects like training efficiency, representational stability, and task-specific performance, such as RoBERTa (Liu et al., 2019). In parallel, compact variants like BERT-tiny and DistilBERT enabled the deployment of transformer architectures in computationally-constrained environments. These lightweight models preserve much of the semantic power of their larger counterparts while offering faster inference and lower memory footprints (Devlin et al., 2019). Crucially, continued research on open-weight models remains vital because they democratize ac-

cess to models, code, and openly documented training corpora, enabling community auditing, reproducible evaluation, and intermediate-layer interpretability (Kapoor et al., 2024; Laurençon et al., 2022).

Conformal prediction (CP) has emerged as a rigorous, distribution-free approach for quantifying uncertainty and improving interpretability in machine learning models. Unlike traditional uncertainty estimation techniques that rely heavily on model-specific internals, CP provides a straightforward yet powerful guarantee: it constructs *prediction sets* that contain the true output with a predefined probability, such as 95%, independent of the model or the underlying data distribution (Angelopoulos & Bates, 2021). This property makes CP especially appealing for deep neural networks and transformer architectures as it provides reliable calibration (Messoudi et al., 2020). Furthermore, CP can adapt to various learning tasks—classification, regression, anomaly detection; through careful design of the nonconformity function, which maps a model prediction and its context to a scalar that reflects uncertainty (Angelopoulos & Bates, 2021). The CONFINE (CONformal INterpretable Explanation) algorithm (Huang et al., 2025) extends CP to neural networks, preserving its theoretical rigor while enhancing its interpretability and robustness.

Building upon CONFINE, this article introduces **CONFIDE** (**CON**formal prediction for **FI**ne-tuned **DE**ep language models), extending conformal prediction techniques to encoder-based transformer architectures. CONFIDE addresses unique transformer challenges, such as layered attention complexities and sequential dependencies, thereby enhancing interpretability and reliability in transformer predictions.

The article makes the following contributions.

- It proposes **CONFIDE**, a conformal prediction framework tailored to fine-tuned language-based transformer models, enabling layer-wise uncertainty calibration and interpretable prediction sets.

- It shows that CONFIDE achieves up to 4.09% absolute accuracy improvement and up to 5.40% higher correct efficiency over softmax-based confidence baselines across GLUE and SuperGLUE benchmarks.

- It demonstrates that CONFIDE outperforms prior uncertainty quantification and interpretability methods, such as NM2 and VanillaNN, on tasks where standard predictors suffer from undercoverage or skewed confidence.

The article is organized as follows. Section 2 dives further into background and related works on model architecture, explainable AI, and CP, including the CONFINE algorithm. Section 3 presents datasets, models, and metrics, followed by the paper's methodology in Section 4. Section 5 presents experimental results, while Section 6 discusses limitations. Section 7 summarizes CONFIDE's findings and use.

## 2 Background and Related Work

In this section, we provide background material and discuss related work.

### 2.1 Transformer Architectures and Interpretability

Transformers revolutionized NLP by replacing sequential recurrence with self-attention, enabling parallel processing and modeling of long-range dependencies for improved contextual representation (Vaswani et al., 2017). The complexity of transformers, however, with multiple layers of self-attention blocks and millions of parameters, introduces interpretability and reliability challenges, as models may exploit superficial dataset cues rather than true semantic structure (Gao et al., 2021; Talebi et al., 2024).

Interpretability (understanding a model's internal logic) and explainability (justifying predictions) are crucial for safe deployment of machine learning models (Leblanc & Germain, 2024).Traditionally, simpler models, such as decision trees, achieve interpretability through intuitive visualizations, like feature importance plots, which reveal the factors influencing each prediction (Quinlan, 1986). In contrast, the multi-head attention layers of transformers entangle representations across tokens, making it difficult to trace how inputs map to outputs. Using built-in decision metrics, such as softmax scores, is also unreliable, as these scores do not reflect the true likelihood of correctness, but instead mirror the model's internal logits (Pearce et al., 2021).

Several interpretability methods have been proposed, including attention visualization (Vig, 2019) and gradient-based attribution (Sundararajan et al., 2017). While these methods provide partial insights, they are often computationally demanding and remain poorly calibrated, especially on out-of-distribution data. Moreover, attention weights are not always reliable indicators of token importance, as studies show that patterns can be redundant or misleading and not directly tied to prediction outcomes (Jain & Wallace, 2019; Wiegreffe & Pinter, 2019). These limitations motivate the need for more principled approaches, such as conformal prediction (CP), which we introduce in the next section.

## 2.2 Conformal Prediction

One method that has had some success is CP. It is a distribution-free framework for uncertainty quantification that offers formal statistical guarantees (Shafer & Vovk, 2008; Lei et al., 2013). By constructing *prediction sets* that contain the true label with a user-defined probability (e.g., 95%), CP ensures that the model's uncertainty estimates are valid regardless of the underlying data distribution or model architecture (Angelopoulos & Bates, 2021; Vovk et al., 2005; Lei et al., 2013). Central to CP is the use of a *nonconformity measure*, which quantifies how unusual or "nonconforming" a test example is relative to a set of calibration examples (Angelopoulos & Bates, 2021). Prediction set sizes, both in CP and other interpretability mechanisms, can be adjusted based on how "confident" a model is in its predictions, quantifying uncertainty (Lei, 2014; Sadinle et al., 2018; Dhillon et al., 2024a). CP methodologies vary in their choice of nonconformity measures, which map model predictions to scalar values that indicate how atypical a prediction is. Common choices include distance-based measures, such as cosine or Euclidean distance, and softmax-based measures (Huang et al., 2025). For classification tasks, Romano et al. (2020) develop conformal predictors that guarantee finite-sample marginal validity while adaptively sizing prediction sets to track instance difficulty, achieving valid and adaptive coverage via a tailored conformity score for unordered labels.

Each metric differs in efficiency and informativeness. For example, 1-nearest neighbor (1NN) in raw input space is cheap but shallow, often yielding overly conservative sets (Papernot & McDaniel, 2018). Deep $k$-nearest neighbors (D$k$NN) leverage hidden-layer activations to align predictions with semantic similarity (Huang et al., 2025), but at high computational and memory cost. In contrast, softmax-based scores are efficient but prone to overconfidence and adversarial vulnerability (Cardenas et al., 2023).

### 2.2.1 CONFINE: CP for Neural Networks

CONFINE, presented by Huang et al. (2025), introduces a novel feature-based nonconformity score that compares the test example to its top-$k$ nearest neighbors in an intermediate representation space (e.g., from a specific network layer). It compares distances to same- vs. different-class neighbors in the embedding space, enabling p-value estimation (discussed later) based on semantic similarity. Thus, CONFINE leverages the *internal* structure of neural networks through representations from a fixed intermediate layer to compute nonconformity scores that more accurately reflect semantic similarity (Huang et al., 2025).

Formally, for each candidate class label $y$, CONFINE calculates the average distance between the test input's embedding and its $k$ nearest neighbors in a calibration set $B$. This yields the following nonconformity score:

$$A_k(B, x, y) = \frac{\min\left\{\frac{1}{k}\sum_{i=1}^{k} \text{CosDist}(f(x), f(x_i)) \mid y_i = y\right\}}{\min\left\{\frac{1}{k}\sum_{i=1}^{k} \text{CosDist}(f(x), f(x_i)) \mid y_i \neq y\right\}}, \tag{1}$$

where $f(x)$ denotes the embedding from a selected layer of the neural network, $(x_i, y_i) \in B$, and $\text{CosDist}(a, b) = 1 - \frac{a \cdot b}{\|a\|\|b\|}$ is the cosine distance.

By fixing the embedding layer (selected via hyperparameter search), CONFINE avoids the prohibitive cost of D$k$NN's multi-layer comparisons while still retaining semantically meaningful structure. In addition, CONFINE supports class-conditional p-value computation, offering improved coverage and label-wise reliability in settings with imbalanced data (Huang et al., 2025).

## 2.3 Conformal Prediction for Transformers

While CONFINE demonstrated that CP can be adapted to deep neural networks, directly applying it to transformers presents unique challenges. Most prior CP approaches suffer from one of three limitations. First, they rely almost exclusively on final-layer embeddings, discarding the rich semantic structure encoded in intermediate layers (Huang et al., 2025). Second, they are typically evaluated only on large-scale models (e.g., BERT or RoBERTa), leaving open the question of whether CP can be effective for lightweight architectures. Third, many implementations reduce nonconformity to simple softmax-based scores, which are efficient but often brittle and poorly calibrated in high-dimensional embedding spaces (Sikar et al., 2025). Geometry-aware choices (e.g., Mahalanobis) and dimension-reduced subspaces are rarely compared systematically, limiting guidance on robustness and interpretability.

Several recent works have begun to address these shortcomings. Giovannotti et al. (2021) apply CP to transformers for paraphrase detection using raw output scores and Mondrian variants. Dey et al. explore inductive CP for text infilling and part-of-speech tagging, achieving finite-sample error control with transformer and BiLSTM embeddings (Dey et al., 2021). Lee et al. extend transformer decoders with quantile-based conformal intervals for time series forecasting, capturing temporal dependencies (Lee et al., 2024). These advances highlight the potential of CP in sequential architectures but also underscore the need for a framework tailored specifically to transformers.

CONFIDE builds on these studies by leveraging intermediate-layer representations to form $k$NN-based nonconformity scores, yielding stable prediction sets across model scales, including lightweight backbones suitable for constrained deployments. It explicitly compares geometry-sensitive metrics (cosine vs. Mahalanobis) and principal component analysis (PCA) to disentangle representation effects on coverage and correct efficiency. Finally, it is evaluated on GLUE and three SuperGLUE tasks, with both marginal and classwise diagnostics, providing a stricter test of calibration and failure modes.

# 3 Datasets and Models

This section introduces our experimental setup to provide grounding for many of CONFIDE's design choices. We discuss datasets used, model choice, and model fine-tuning.

## 3.1 GLUE and SuperGLUE Benchmarks

To evaluate CONFIDE, we use tasks from two widely adopted natural language understanding benchmarks—GLUE (Wang et al., 2019) and SuperGLUE (Wang et al., 2020). Conformal predictions attention to problems labeled as classification (excluding regression or span-level predictions such as STS-B). The resulting suite spans binary question answering through multi-class natural language inference and serves as a rigorous testbed for uncertainty quantification: for instance, BoolQ is a binary QA dataset drawn from naturally occurring queries with ambiguous phrasing and occasional label noise, while CB (Commitment-Bank) is a three-class NLI task with limited training data, making it valuable for assessing calibration under data scarcity. A full list of included tasks and their characteristics is provided in Appendix A.

## 3.2 Encoder-Based Transformer Models

We evaluate CP using BERT and RoBERTa, as they produce sentence-level representations suited to classification, expose per-layer hidden states needed for CONFIDE's layer-conditioned nonconformity scores, and are available as reproducible, open-source checkpoints. In contrast, decoder-only or encoder–decoder models (e.g., GPT, T5) target generation and add decoding-specific complexity that is orthogonal to our classification focus (Devlin et al., 2019; Liu et al., 2019; Raffel et al., 2023). To study scaling effects without confounding architecture, we span four pretrained variants.

- `BERT-tiny`: 2 layers, ~4M params

- `BERT-small`: 4 layers, ~29M params

- `RoBERTa-base`: 12 layers, $\sim$125M params

- `RoBERTa-large`: 24 layers, $\sim$355M params

### 3.3 Fine-tuning Protocol

We apply a single HuggingFace `Trainer`-based recipe to all model-task pairs to avoid confounds and keep the prediction backbone fixed for CONFIDE.

- **Backbone.** Load a pretrained checkpoint and replace the classification head to match task labels; the final checkpoint is chosen by *best validation accuracy.*

- **Inputs.** Task-aware tokenization: sentence-pair formatting for BoolQ/RTE/CB; MultiRC treated as binary over (question, answer) pairs. Pad/truncate to 512 tokens.

- **Optimization.** AdamW (learning rate $5\times10^{-5}$, weight decay 0.01) with linear warmup (first 500 steps); train up to 10 epochs with early stopping on validation accuracy (typically 3–5) (Howard & Ruder, 2018).

- **Metrics.** Accuracy is the selection metric; precision/recall/F1 are logged (macro for multi-class, binary otherwise).

All CONFIDE evaluations use the selected checkpoint without further tuning, ensuring comparability across datasets and architectures.

## 4 Methodology

This section introduces CONFIDE, built atop the Confine framework, for transformer-based language models. We outline the Confine algorithm, differentiate the CONFIDE algorithm, and detail design decisions.

### 4.1 Assumptions and Definitions

The CONFINE algorithm provides prediction sets that are calibrated under the exchangeability assumption and leverages the internal structure of neural networks to compute semantically meaningful nonconformity scores.

- **Exchangeability assumption.** Let $(X_1, Y_1), \ldots, (X_n, Y_n)$ be random pairs in $\mathcal{X} \times \mathcal{Y}$. They are *exchangeable* if their joint law is invariant under permutations: for any permutation $\pi$ of $\{1, \ldots, n\}$

$$f(X_1, Y_1, \ldots, X_n, Y_n) \stackrel{d}{=} (X_{\pi(1)}, Y_{\pi(1)}, \ldots, X_{\pi(n)}, Y_{\pi(n)}). \tag{2}$$

- **P-values and prediction sets (single definition)**: For a test input $x_{l+1}$ and candidate label $y_j$, define the nonconformity $\alpha_{l+1} = A(B, x_{l+1}, y_j)$. Let $\{\alpha_i(y_j)\}_{i \in \mathcal{C}_{y_j}}$ be the calibration scores for label $y_j$ (pooled if marginal, same-label if Mondrian). The conformal $p$-value and prediction set are

$$p(y_j) \;=\; \frac{\#\{i \in \mathcal{C}_{y_j} : \alpha_i(y_j) \geq \alpha_{l+1}\} + 1}{|\mathcal{C}_{y_j}| + 1}, \qquad \Gamma^\varepsilon(x) \;=\; \{\, y_j \in \mathcal{Y} \mid p(y_j) > \varepsilon \,\}. \tag{3}$$

We use this definition throughout. "Pooled" (marginal) uses all calibration points for every $y_j$; "class-conditional" (Mondrian) uses only calibration points with label $y_j$. We never filter the calibration set by model correctness; doing so would break exchangeability and invalidate finite-sample guarantees.

- **Marginal coverage guarantee**: Under exchangeability of $(Z_i)_{i \in \mathcal{C}} \cup \{Z_{n_{\text{cal}}+1}\}$,

$$\Pr\big[\, Y \in \Gamma^\varepsilon(X) \,\big] \;\geq\; 1 - \varepsilon. \tag{4}$$

  This is the standard (marginal) guarantee: The true label appears in the set with probability at least $1 - \varepsilon$, averaged over the test distribution (with randomness in the calibration split).

- **Class-conditional coverage**: A per-class guarantee holds when calibration is performed within each class:
$$\Pr\big[\, Y \in \Gamma^\varepsilon(X) \,\big|\, Y = y \,\big] \;\geq\; 1 - \varepsilon, \qquad \forall\, y \in \mathcal{Y}. \tag{5}$$

  This ensures each class attains the nominal coverage, not just the average over classes.

- **Credibility and confidence**:
$$\text{Credibility} \;=\; \max_{y_j \in \mathcal{Y}} p(y_j), \qquad \text{Confidence} \;=\; 1 \;-\; \max_{y_j \neq y^*} p(y_j), \tag{6}$$

  where $y^* = \arg\max_{y_j} p(y_j)$. Credibility reflects how well the top label conforms; confidence measures its separation from the rest.

- **Efficiency and correct efficiency**: The average set size
$$\mathbb{E}\big[\, |\Gamma^\varepsilon(x)| \,\big] \tag{7}$$

  measures *efficiency* **(lower is better)**. Because one can appear "efficient" by shrinking sets while omitting the truth, we also report a normalized, *correct compactness score*, called correct efficiency, restricted to correctly covering sets:

$$\text{cEff} \;=\; 1 \;-\; \mathbb{E}\Big[ \tfrac{|\Gamma^\varepsilon(x)|}{|\mathcal{Y}|} \;\Big|\; Y \in \Gamma^\varepsilon(x) \Big] \;\in [0, 1] \quad \text{(higher is better)}.$$

  In all results we report, coverage, efficiency (lower is better), and correct efficiency (higher is better) together to prevent the "manipulation" where tiny sets miss the true label (Huang et al., 2025; Vovk et al., 2016).

- **Transformer representation modes (Flattened vs. Attention).** For a chosen layer $\ell$ with $H_\ell(x) \in \mathbb{R}^{T \times d}$, we form $h(x)$ by either *Attention (CLS)*: $h(x) = H_\ell(x)_{[\texttt{CLS}]} \in \mathbb{R}^d$, or *Flattened*: $h(x) = \text{vec}(H_\ell(x)) \in \mathbb{R}^{Td}$.

- **Layer selection.** The layer index $\ell$ is a hyperparameter; once chosen, the same $\ell$ is used for training/reference, calibration, and test extraction to keep $h(x)$ comparable. If the final logits/softmax layer is used, we optionally apply temperature scaling with parameter $T$ (i.e., $\text{softmax}(z/T)$).

- **Reference pools vs. calibration.** During reference construction, we retain only correctly predicted training points ($\hat{y}_i = y_i$) in the per-class $k$NN pools. The *calibration* set is never filtered by correctness, preserving exchangeability and finite-sample validity.

## 4.2 The CONFIDE Algorithm

CONFIDE introduces two key adaptations: (1) a representation mode selector to aggregate token-level embeddings and (2) an optional dimensionality reduction step and distance-metric modifier for handling high-dimensional transformer outputs. The full pipeline is detailed in Algorithm 1.

As noted, for a specified layer $\ell$, CONFIDE extracts representations using either a *flattened* mode, which reshapes the full token matrix into a vector, or an *attention* mode, which selects the [CLS] token embedding. For comparisons between modes, we hold all other hyperparamters constant.

Importantly, we use *numeric layer indices* ($\ell$) to refer to positions in the backbone's module graph. For reproducibility, we also map each $\ell$ to its canonical transformer submodule (e.g., `encoder.layer.10.output`); see Appendix B.2. Fine-tuning changes weights but not depth or ordering; hence, a given $\ell$ maps to the same submodule across datasets for a fixed backbone.

## Step 1: Representation Extraction

From a chosen encoder layer $\ell$, we either build a single [CLS] embedding—one compact vector per example *(attention)*, or concatenate all token embeddings from that layer into one long vector *(flattened)*. Flattened can be more compute intensive because the feature dimension scales with $T_d$; we therefore default to *attention* for most experiments.

---

**Algorithm 1** The CONFIDE Algorithm

---

**Input:** Labeled training dataset $(X, Y)$; model $\mathcal{M}$; significance level $\epsilon$; test input $z$; layer $\ell$; number of neighbors $k$; representation mode; PCA flag; distance metric

**Step 1: Extract Representations from Calibration Data**

  **foreach** *training example* $(x_i, y_i)$ **do**
  
   Run $\mathcal{M}(x_i)$ to extract $H_i$ from layer $\ell$ **if** *mode == `Flattened`* **then**
    $h_i \leftarrow$ `reshape`$(H_i, [1, \text{seq\_len} \cdot \text{hidden\_dim}])$ ;                    // Flatten full matrix
   **else if** *mode == `Attention`* **then**
    $h_i \leftarrow H_i[\text{CLS}]$ ;                                            // Use [CLS] token
   **if** $\hat{y}_i = y_i$ **then**
    Store $h_i$ in pool for class $y_i$

**Step 2: Apply PCA (Optional)**

  **if** *PCA enabled* **then**
   Fit PCA on $\{h_i\}$ to retain 95% variance and transform all vectors

**Step 3: Fit $k$NN Models per Class**

  **foreach** *class* $c \in \mathcal{Y}$ **do**
   Fit $k$NN on $\{h_i \mid y_i = c\}$ using selected distance metric

**Step 4: Calibration Nonconformity Scoring**

  **foreach** *calibration example* $(x, y)$ **do**
   Extract $h(x)$ as above; Compute $A_k(x, y) = \frac{\text{avg dist to class } y}{\text{avg dist to other classes}}$; Store score in $\alpha$

**Step 5: Compute Test Prediction Sets**

  **foreach** *test input* $x$ **do**
   **foreach** *label* $y \in \mathcal{Y}$ **do**
    Compute $A_k(x, y)$ and p-value:    $p_y(x) = \frac{1 + \#\{\alpha_i \geq A_k(x,y)\}}{1 + n_{\text{cal}}}$
   Output prediction set: $\Gamma^\epsilon(x) = \{ y \in \mathcal{Y} \mid p_y(x) > \epsilon \}$

---

## Step 2: Apply PCA (Optional)

To mitigate the curse of dimensionality and improve distance metric robustness, PCA may be optionally applied. If enabled, PCA reduces the dimensionality of all $h_i$ vectors while retaining at least 95% of the variance. This transformation is also applied to calibration and test embeddings.

## Step 3: Fit Class-Conditional $k$NN Models

For each class $c$, we build a $k$NN index on the layer-$\ell$ embeddings of *correctly predicted training examples* from that class (the "positives," $R_c$) and use the union of all other classes as the "negatives" $\bigcup_{c' \neq c} R_{c'}$. At test time, for each candidate label $y$, we query the $k$ nearest positives from $R_y$ and the $k$ nearest negatives from $\bigcup_{c \neq y} R_c$ using the chosen distance (cosine or Mahalanobis), and define nonconformity from the ratio of their average distances. We retain only correctly predicted *training* points for reference (to denoise the neighborhoods), but *do not* filter calibration points.

## Step 4: Calibration Nonconformity Scoring

For each calibration point $(x, y)$, a nonconformity score $A_k(x, y)$ is computed using the same embedding extraction and PCA pipeline. These scores quantify how typical a point $x$ is for its true class $y$. They are stored and later used to derive p-values for each candidate class during prediction.

**Step 5: Test-time Prediction Set**

At test time, for each input $x$, a p-value is computed for every possible class $y \in \mathcal{Y}$, as well as a prediction set. This set guarantees marginal coverage $1 - \epsilon$ under the exchangeability assumption. The size and accuracy of this set depend on the quality of extracted embeddings and chosen distance metric.

We evaluate CONFIDE variants using three core metrics: accuracy, correct efficiency, and coverage (Huang et al., 2025). We adopt baselines that span two standard CP families: distance-based (VanillaNN) and softmax-based (NM1/NM2). As was done in CONFINE, these are chosen as they are training-free and parameter-minimal; hence, we can isolate representation effects without confounding results with $k$-tuning or additional fitting (Vovk et al., 2005; Huang et al., 2025). Specifically, NM1/NM2 are canonical softmax-derived nonconformity scores for classification (inverse probability and margin), widely used in the CP literature and tutorials as fast, model-agnostic baselines that mirror probability-thresholding practice (Vovk et al., 2005; Angelopoulos & Bates, 2021).

## 4.3 Justifying Design Choices

Building on the CONFIDE variants in Algorithm 1, we summarize the design axes that most affect performance and point to where each comparison is evaluated.

**Distance metric.** We support cosine and Mahalanobis distances in the $k$NN scoring. Cosine distance is defined as $\mathrm{Dist}_{\cos}(u, v) = 1 - \frac{u^\top v}{\|u\| \, \|v\|}$, while Mahalanobis distance is defined as $\mathrm{Dist}_{\mathrm{Mah}}(x, \mu) = \sqrt{(x - \mu)^\top \Sigma^{-1}(x - \mu)}$. Cosine distance is scale-invariant and robust in high dimensions; Mahalanobis distance captures feature correlations and can improve class discrimination, but is sensitive to ill-conditioned $\Sigma$ (Moutafis et al., 2017).

**Dimensionality reduction.** To mitigate high-dimensional effects and stabilize Mahalanobis distance, we optionally apply PCA to centered activations and retain components explaining $\geq 95\%$ variance before fitting/querying $k$NN; the same transform is applied at calibration and test time (Jolliffe & Cadima, 2016).

**Hyperparameters and search.** We tune three hyperparameters per (model, dataset): chosen *Layer*, the number of neighbors $k$, and the temperature $T$ for any softmax-based baselines. We conduct a grid search over appropriate ranges for each hyperparameter within every (model, dataset) configuration, ensuring that each CONFIDE variant is tuned for both predictive accuracy and valid coverage. The values for $k$ and $T$ follow those used in the original CONFINE framework (Huang et al., 2025), facilitating a direct comparison. For the *layer* parameter, we select a diverse set of layers spanning the early, middle, and late stages of the encoder. These layers are chosen to reflect different architectural roles, ranging from multi-head attention blocks to linear feedforward components, to capture the progression of semantic abstraction across the model. The full list of layers chosen can be found in Appendix B.3.

These design axes are evaluated across all four transformer models and eleven tasks. Each configuration undergoes comprehensive hyperparameter tuning to ensure fair and reproducible comparisons. Our results, presented in the following section, highlight the impact of these design choices on both prediction set coverage and efficiency.

# 5 Experimental Results

Across models, CONFIDE improves correct efficiency while preserving accuracy; however, we frequently observe substantial undercoverage on hard or minority classes (e.g., CoLA "unacceptable," BoolQ "false"), and no approach reaches the nominal $1 - \varepsilon$ target when the exchangeability condition is violated. Classwise calibration mitigates but does not eliminate these failures. We, therefore, report classwise diagnostics alongside aggregate metrics and emphasize use with complementary safeguards.

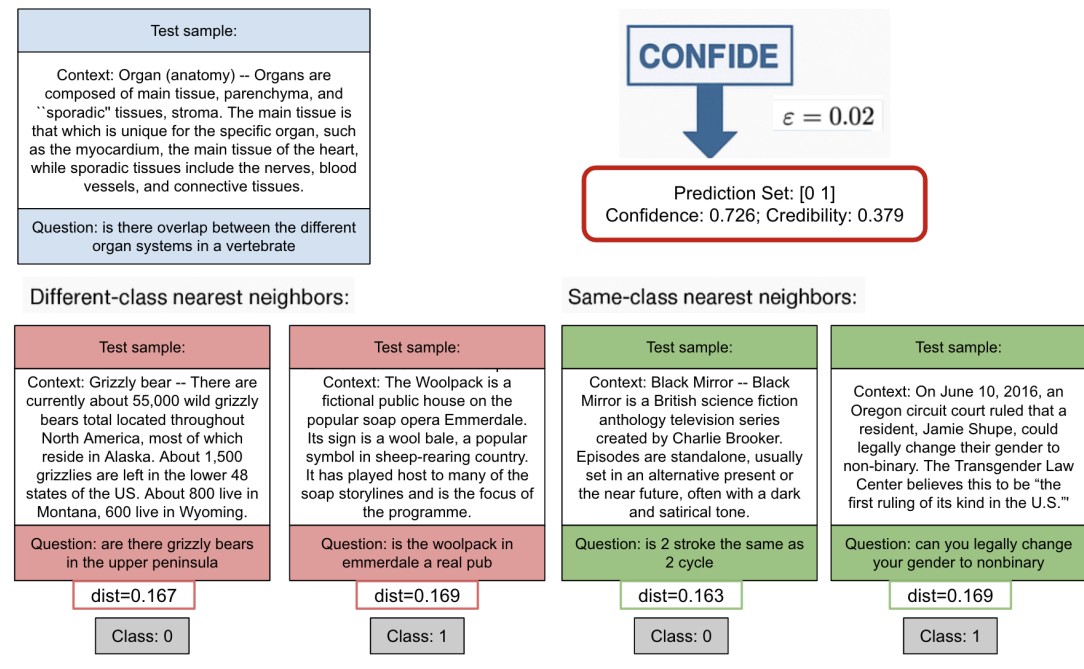

Figure 1: Failure case on BoolQ (BERT-tiny). CONFIDE fails to distinguish between same-class and different-class neighbors, resulting in a high-uncertainty prediction.

## 5.1 CONFIDE Provides Interpretability

CONFIDE offers interpretable insights on individual test cases. Fig. 1 demonstrates how an end-user can inspect predictions by visualizing the predicted set, associated confidence metrics, and the structure of nearest neighbors. We use "interpretable" to mean case-level *representation-space evidence*: for each candidate label, we surface nearest *supporting* (same-class) and *contradicting* (other-class) neighbors with their distances. Hence, a user can audit whether the embedding provides a label-separating structure; this is a plausibility/consistency diagnostic, not a causal explanation. In contrast to image-based models, where neighbor similarity reflects visual resemblance, language models encode similarity based on shared semantic structures, syntactic roles, or contextual usage. Thus, nearby neighbors in CONFIDE often reflect questions or passages with comparable phrasing, topic domains, or logical structure, which can be less intuitive than for vision tasks.

Fig. 1 presents a failure case on the BoolQ dataset. The model outputs a full prediction set $\{0, 1\}$, with an associated credibility of only 0.379, signaling substantial uncertainty: the model is unable to confidently distinguish between the two classes. Importantly, the average distances to same-class and different-class neighbors are nearly identical (0.16–0.17), indicating that the embedding space lacks meaningful class separation for this example. Fittingly, the test sample is a medically themed question; an end user would see that the model has low representational certainty and could defer to a domain expert, such as a doctor, for validation. As seen by both the different-class and same-class neighbors, no similar sample appears to exist in the dataset, hence, the model is not able to confidently predict the right class. In comparison, a successful CONFIDE prediction would return a singleton prediction with high *confidence* and *credibility*, which we share in Appendix D, Fig. 16.

## 5.2 Performance on Resource-Constrained Transformer Models

We first evaluate CONFIDE on compact transformer models: BERT-tiny and BERT-small. Full results are reported in Tables 1–3. We report two metrics: **CONFIDE-A** shares the best performing hyperparameters to maximize top-1 accuracy, which is appropriate when a single decision must be issued. **CONFIDE-C**

Table 1: Performance on GLUE benchmarks using BERT-tiny: CoLA, MNLI, MRPC, and QNLI. Hyperparameters used can be found in Appendix B.1

| Method | CoLA | | MNLI | | MRPC | | QNLI | |
|---|---|---|---|---|---|---|---|---|
| | Test Acc | Top Corr Eff | Test Acc | Top Corr Eff | Test Acc | Top Corr Eff | Test Acc | Top Corr Eff |
| Original NN | 0.5858 | – | 0.6949 | – | 0.7010 | – | 0.7895 | – |
| 1-Nearest Neighbor | 0.5791 | 0.5762 | 0.3403 | 0.3254 | 0.6103 | 0.6103 | 0.5067 | 0.5050 |
| NM (2) | 0.5858 | 0.5724 | 0.6949 | **0.6936** | 0.7010 | 0.7010 | 0.7895 | 0.7878 |
| CONFIDE-A (ours) | **0.6903** | **0.6903** | **0.6982** | 0.6907 | **0.7304** | 0.7181 | **0.8157** | 0.8104 |
| CONFIDE-C (ours) | **0.6903** | **0.6903** | 0.6975 | 0.6932 | 0.7279 | **0.7279** | 0.8131 | **0.8129** |

Table 2: Performance on GLUE benchmarks using BERT-tiny: QQP, RTE, SST-2, and WNLI.

| Method | QQP | | RTE | | SST-2 | | WNLI | |
|---|---|---|---|---|---|---|---|---|
| | Test Acc | Top Corr Eff | Test Acc | Top Corr Eff | Test Acc | Top Corr Eff | Test Acc | Top Corr Eff |
| Original NN | 0.8116 | – | 0.5668 | – | 0.8303 | – | 0.5634 | – |
| 1-Nearest Neighbor | 0.6423 | 0.6391 | 0.4693 | 0.4549 | 0.5011 | 0.4794 | 0.2394 | 0.2394 |
| NM (2) | 0.8214 | 0.8073 | 0.5668 | 0.5668 | 0.8303 | 0.8280 | 0.5634 | 0.5634 |
| CONFIDE-A (ours) | **0.8566** | **0.8530** | **0.5957** | 0.5848 | **0.8372** | 0.8326 | **0.8360** | 0.8245 |
| CONFIDE-C (ours) | **0.8566** | **0.8530** | 0.5884 | **0.5884** | 0.8349 | **0.8349** | 0.8326 | **0.8280** |

instead maximizes *correct efficiency*, favoring configurations that produce smaller, more decisive sets when they are correct, reducing ambiguity and review cost.

Both models benefit significantly from CONFIDE's nonconformity-based calibration, improving upon the NM1/NM2 baselines by flexibly selecting semantically rich internal layers, enabling it to outperform strong baselines across several benchmarks. For example, on SST-2, CONFIDE-C boosts top-1 correct efficiency to **0.8911** on BERT-small and **0.8349** on BERT-tinym compared to just **0.8830** and **0.8280**, respectively, under NM1/NM2. On QQP, BERT-small improves from **0.8739** (NM) to **0.8846** (CONFIDE-C), while BERT-tiny improves from **0.8073** to **0.8530**. MRPC shows similar trends: on BERT-small, CONFIDE-C achieves **0.7574** correct efficiency, compared to **0.7034** with a previous NM; BERT-tiny moves from **0.7010** to **0.7279**.

An important trend across both models is that CONFIDE achieves its strongest performance not from the final softmax layer but from intermediate transformer layers. Based on our grid search (see Appendix B.3), optimal configurations often use mid-layer embeddings (Appendix B.1). For **BERT-tiny**, top-performing runs often used $l{=}16$ (*encoder.layer.0.attention.output*). For **BERT-small**, strong results are frequently found at $l{=}27$ (*encoder.layer.1*) and $l{=}34$ (*encoder.layer.1.attention.output*).

These findings suggest that early-layer representations capture richer features that are especially amenable to class-conditional distance metrics. Specifically, because these embeddings retain a nuanced semantic structure without being overly aligned to the model's final output, they allow CONFIDE to more effectively compare a test input's proximity to examples from the same class versus different classes.

Importantly, a strong model still matters: BERT-small > BERT-tiny. Across nearly every task, BERT-small consistently outperforms BERT-tiny in both test accuracy and correct efficiency. On MRPC, QNLI, and QQP, the performance gap is clear: BERT-small with CONFIDE-C achieves correct efficiencies of **0.7574**, **0.8697**, and **0.8846** respectively, whereas BERT-tiny peaks at **0.7279**, **0.8129**, and **0.8530**. Even modest increases in model depth (from 2 layers to 4) enable CONFIDE to form tighter, more accurate prediction sets. Additional results are presented in Appendix D.

Despite CONFIDE's consistent improvements, both models still struggle on several particularly difficult tasks — especially CB, WNLI, BoolQ, and MultiRC — where ambiguity, low supervision, or multi-label structure

Table 3: Performance on SuperGLUE benchmarks using BERT-small: BoolQ, CB, and MultiRC.

| Method | BoolQ | | CB | | MultiRC | |
|---|---|---|---|---|---|---|
| | Test Acc | Top Corr Eff | Test Acc | Top Corr Eff | Test Acc | Top Corr Eff |
| Original NN | 0.7275 | – | 0.6250 | – | **0.6640** | – |
| 1-Nearest Neighbor | 0.5602 | 0.5590 | 0.4286 | 0.4286 | 0.5192 | 0.4792 |
| NM1/NM2 | 0.7275 | 0.6269 | 0.6250 | 0.6250 | **0.6640** | **0.6621** |
| CONFIDE-A (ours) | **0.7297** | 0.6104 | **0.7143** | 0.6786 | 0.6625 | 0.6557 |
| CONFIDE-C (ours) | 0.7272 | **0.7245** | **0.7143** | **0.7143** | 0.6599 | 0.6582 |

Table 4: Performance on GLUE benchmarks using RoBERTa-base: CoLA, MNLI, MRPC, and QNLI.

| Method | CoLA | | MNLI | | MRPC | | QNLI | |
|---|---|---|---|---|---|---|---|---|
| | Test Acc | Top Corr Eff | Test Acc | Top Corr Eff | Test Acc | Top Corr Eff | Test Acc | Top Corr Eff |
| Original NN | 0.8408 | – | – | – | 0.8824 | – | 0.9231 | – |
| 1-Nearest Neighbor | 0.6069 | 0.6021 | – | – | 0.5564 | 0.5490 | – | – |
| NM1/NM2 | 0.8408 | 0.8370 | **0.8608** | 0.8497 | 0.8824 | 0.8627 | 0.9231 | 0.9098 |
| CONFIDE-A (ours) | **0.8495** | 0.8092 | **0.8608** | 0.8497 | **0.8848** | 0.8725 | **0.9235** | 0.9087 |
| CONFIDE-C (ours) | **0.8495** | **0.8495** | 0.8588 | **0.8512** | **0.8848** | **0.8848** | **0.9235** | **0.9217** |

present major challenges. In these cases, even the best CONFIDE configurations offer only modest gains, as the base model itself often fails to learn a strong decision boundary. In the most favorable configurations, CONFIDE improves test accuracy by up to 4.09% for BERT-tiny and 2.69% for BERT-small over the best baseline methods. These results highlight that while conformal calibration can meaningfully improve reliability, overall performance remains bottle necked by base model capacity.

### 5.3 Performance on Larger Models

Similarly, across both RoBERTa variants, CONFIDE consistently improves top-1 correct efficiency relative to softmax-based baselines like NM1 and NM2, while preserving or slightly improving classification accuracy. For example, on MRPC, CONFIDE-C lifts correct efficiency from **0.8627** to **0.8848** on RoBERTa-base, and from **0.8578** to **0.8652** on RoBERTa-large (Tables 4, 7). Similar 1–2 percentage point gains are observed on RTE, SST-2, and CoLA: on RTE, CONFIDE-A and CONFIDE-C increase top-1 efficiency from **0.7112** (NM2) to **0.7256** and **0.7292**, respectively (Table 5), while CoLA improves from **0.8370** to **0.8495** (Table 4). Even on high-performing tasks like MNLI, CONFIDE-C yields a marginal improvement from **0.8497** to **0.8512**. Crucially, these calibration benefits are not achieved at the cost of accuracy. On RTE, CONFIDE-C increases accuracy from **0.7220** (NM2) to **0.7329**, with CONFIDE-A reaching **0.7365** — the highest among all variants (Table 5).

RoBERTa-large exhibits a consistently narrower gap between test accuracy and correct efficiency, even before applying CONFIDE. However, CONFIDE still meaningfully refines this calibration. On MNLI and RTE, RoBERTa-large under CONFIDE-A achieves **0.8437** efficiency on CoLA and **0.7978** on RTE. Larger models also enable smaller, more compact prediction sets. For example, on the RTE benchmark, the average prediction set size among correct predictions is **1.84** for RoBERTa-base, compared to **1.99** for BERT-tiny (Dhillon et al., 2024b).

From a practical standpoint, CONFIDE is restricted by being resource-intensive. On long-context datasets (BoolQ, MultiRC), RoBERTa-large configurations that used flattened token representations and full pairwise distance computations exceeded our GPU memory budget. We, therefore, report RoBERTa-large results only for configurations that fit within memory constraints; resource profiles (peak memory, batch sizes, run times) are presented in Appendix A, and all completed runs are summarized in Table 7.

Table 5: Performance on GLUE benchmarks using RoBERTa-base: QQP, RTE, SST-2, and WNLI.

| Method | QQP | | RTE | | SST-2 | | WNLI | |
|---|---|---|---|---|---|---|---|---|
| | Test Acc | Top Corr Eff | Test Acc | Top Corr Eff | Test Acc | Top Corr Eff | Test Acc | Top Corr Eff |
| Original NN | 0.9069 | – | 0.7220 | – | 0.9323 | – | 0.4366 | – |
| 1-Nearest Neighbor | – | – | 0.4693 | 0.4657 | 0.5092 | 0.5034 | 0.1972 | 0.1831 |
| NM1/NM2 | 0.9069 | **0.8961** | 0.7220 | 0.7112 | 0.9323 | 0.9255 | 0.4366 | 0.4366 |
| CONFIDE-A (ours) | **0.9075** | 0.8955 | **0.7365** | 0.7256 | **0.9346** | 0.9289 | – | – |
| CONFIDE-C (ours) | 0.9070 | **0.8961** | 0.7329 | **0.7292** | 0.9323 | **0.9323** | – | – |

Table 6: Performance on SuperGLUE benchmarks using RoBERTa-base: BoolQ, CB, and MultiRC.

| Method | BoolQ | | CB | | MultiRC | |
|---|---|---|---|---|---|---|
| | Test Acc | Top Corr Eff | Test Acc | Top Corr Eff | Test Acc | Top Corr Eff |
| Original NN | 0.8092 | – | 0.6786 | – | 0.7473 | – |
| 1-Nearest Neighbor | – | – | 0.4464 | 0.4286 | – | – |
| NM1/NM2 | 0.8092 | 0.7654 | 0.6786 | 0.6607 | **0.7473** | **0.7471** |
| CONFIDE-A (ours) | **0.8095** | 0.5483 | **0.7500** | **0.7500** | 0.4299 | 0.4292 |
| CONFIDE-C (ours) | 0.8089 | **0.8089** | **0.7500** | **0.7500** | 0.4299 | 0.4292 |

## 5.4 Classwise and Aggregate Validity of CONFIDE on BERT-tiny

A central goal of conformal prediction is to guarantee that the true label is included in the model's prediction set with high probability—typically at least $1 - \varepsilon$, where $\varepsilon$ is the target error rate. This is known as *marginal validity* and is achieved when the average coverage of the model's prediction sets exceeds this threshold. However, as emphasized in CONFINE (Huang et al., 2025), marginal validity alone does not ensure equitable performance across all classes. In many applications, particularly high-stakes ones like healthcare, what matters more is *class-conditional validity*: the guarantee that each individual class receives its own calibrated coverage. We analyze both marginal and classwise coverage on BERT-tiny using CONFIDE, focusing on three representative datasets: CoLA, MNLI, and BoolQ. All models are evaluated using attention representations, with all other metrics varied.

### 5.4.1 Comparative Results

CONFIDE's classwise behavior reveals both strengths and limitations in calibration quality across tasks. CoLA presents a failure case. As shown in Fig. 2, the aggregate coverage curve slightly underperforms, falling below the $1 - \varepsilon$ diagonal, with correct efficiency peaking briefly before declining. Classwise analysis reveals the issue: While the "acceptable" class is overcovered, the "unacceptable" class is severely undercovered, with coverage dropping to zero for all $\varepsilon > 0.25$.

BoolQ initially appears valid at the aggregate level, but classwise curves tell a different story (Fig. 3). The model heavily overpredicts the "true" class, assigning it to 906 of 1,237 "false" examples. As a result, the "false" class suffers from poor accuracy (26.8%) and low coverage, while "true" examples enjoy inflated metrics. Confidence and credibility are also skewed, averaging 0.894 vs. 0.865 (confidence) and 0.645 vs. 0.576 (credibility) for "true" and "false," respectively. This reflects a representational collapse: inputs from both classes are embedded too similarly, leading to overconfident, incorrect predictions.

These results underscore a critical limitation of marginal coverage: **Improvements in accuracy and correct efficiency do not guarantee per-class calibration.** When one class dominates model predictions, marginal metrics may appear strong even as minority classes are systematically misrepresented.

Table 7: Performance on GLUE and SuperGLUE benchmarks using RoBERTa-large: CoLA, MRPC, RTE, and WNLI.

| Method | CoLA | | MRPC | | RTE | | WNLI | |
|---|---|---|---|---|---|---|---|---|
| | Test Acc | Top Corr Eff | Test Acc | Top Corr Eff | Test Acc | Top Corr Eff | Test Acc | Top Corr Eff |
| Original NN | 0.8418 | – | 0.8578 | – | 0.8087 | – | **0.5634** | – |
| 1-Nearest Neighbor | 0.6069 | 0.6021 | 0.5564 | 0.5490 | 0.4693 | 0.4657 | 0.1972 | 0.1831 |
| NM1/NM2 | 0.8418 | 0.8293 | 0.8578 | 0.8578 | 0.8087 | **0.8087** | 0.5634 | 0.5493 |
| CONFIDE-A (ours) | **0.8466** | **0.8437** | **0.8710** | 0.8578 | **0.8123** | 0.7978 | 0.5634 | 0.5493 |
| CONFIDE-C (ours) | **0.8466** | **0.8437** | 0.8652 | **0.8652** | 0.8051 | 0.8051 | 0.5493 | 0.5493 |

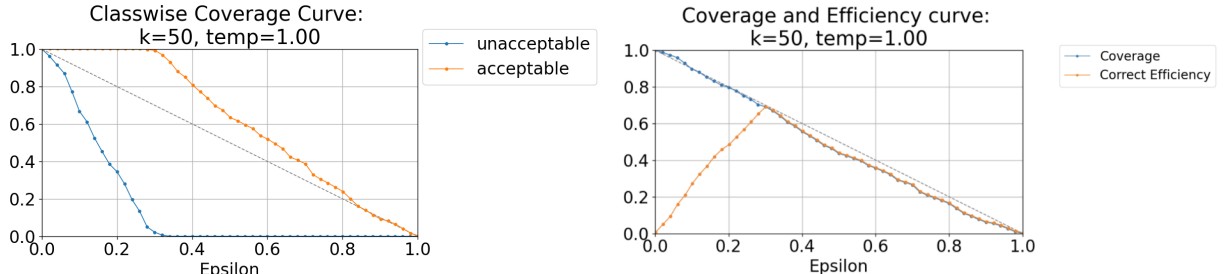

Figure 2: Classwise and aggregate coverage curves for BERT-tiny on CoLA. The false class severely undercovers, violating conformal validity.

### 5.4.2 Classwise Conformal Prediction Improves, but Does Not Solve, Failures of Conformal Validity in BERT-tiny

Classwise adjustment can be valuable when the base model exhibits strong class-specific bias, as it attempts to re-balance coverage in a targeted manner. In BoolQ, applying classwise calibration substantially improves coverage for the minority "false" class, and both classes begin to track more closely along the ideal coverage line (Fig. 4). However, some under-coverage remains, especially under the ideal line for low $\varepsilon$, suggesting residual bias. This suggests that classwise calibration is a useful but partial solution. We also share CoLA graphs (Fig. 5), which show similar improvements but lingering asymmetry. Fully resolving these failures may require further adjustments, such as class reweighting, further temperature scaling, or more expressive representations. Similar results are obtained for BERT-small, included in the appendix.

### 5.5 Calibration Robustness in RoBERTa Models

We similarly analyze CONFIDE's performance under both non-classwise and classwise calibration across CoLA, BoolQ, and CB for the larger models, with additional results shared in Appendix D. These experiments reveal that larger models can produce more stable and exchangeable embeddings but that calibration quality still varies by dataset and class distribution.

**RoBERTa-base: Poor baseline calibration, strong classwise gains.** On CoLA, RoBERTa-base again has poor calibration without classwise correction. As shown in Fig. 6, the aggregate coverage curve is below the $1 - \varepsilon$ diagonal, and the correct efficiency is not tight, indicating that most prediction sets are large and contain incorrect classes. Here, however, enabling classwise calibration on CoLA yields strong improvements, even above the diagonal curve. Even when the embeddings are already well-structured, classwise correction can become a *fine-tuning tool*. We also note the continued poor correct efficiency performance, indicating that CoLA remains a difficult dataset for even the larger models to correctly predict. Further results for comparative classwise metrics are presented in Appendix D.

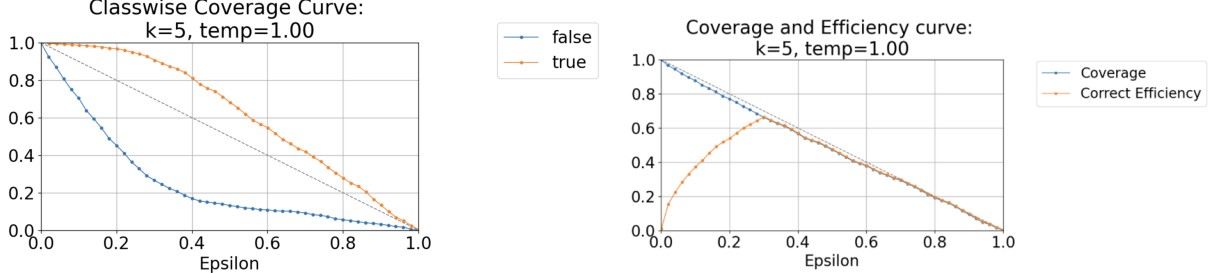

Figure 3: Classwise and aggregate coverage curves for BERT-tiny on BoolQ (non-classwise). The false class shows massive undercoverage due to biased calibration, despite almost diagonal aggregated statistics.

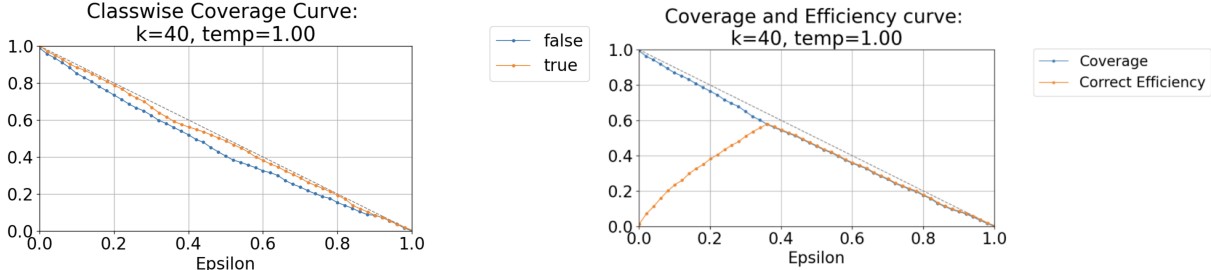

Figure 4: Coverage curves on BoolQ using classwise calibration. The classwise method reduces undercoverage but does not fully resolve it.

This behavior likely reflects instability in the nonconformity score distribution for ambiguous examples. At higher $\varepsilon$ thresholds, CONFIDE is expected to tolerate more errors by assigning lower nonconformity scores to uncertain examples. However, in tasks like BoolQ, where many questions may not be easily distinguishable as "true" or "false" without fine-grained reasoning, the model may have difficulty ranking examples consistently.

RoBERTa-large similarly can show strong calibration on structured datasets, but struggles under label ambiguity. On the CB dataset (Fig. 7), entailment and contradiction maintain reasonable classwise coverage curves, but the "neutral" class fails, remaining below 0.5 across all $\varepsilon$. This is likely due to two factors: the semantic vagueness of the neutral class, and its very low representation in the dataset. Even with high-capacity models, these challenges cannot be resolved purely by scale. The failure of RoBERTa-large on CB highlights the fact that large model size alone is insufficient for tackling calibrated uncertainty under semantic ambiguity and class sparsity. More examples of RoBERTa-large performance are presented in Appendix D.

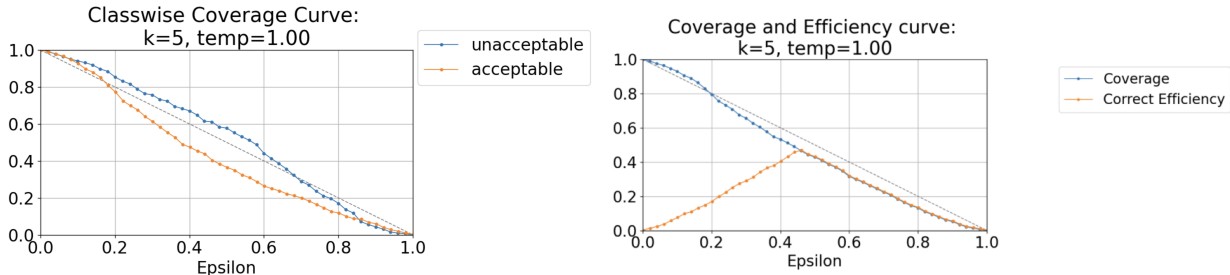

Figure 5: Coverage curves on CoLA using classwise calibration.

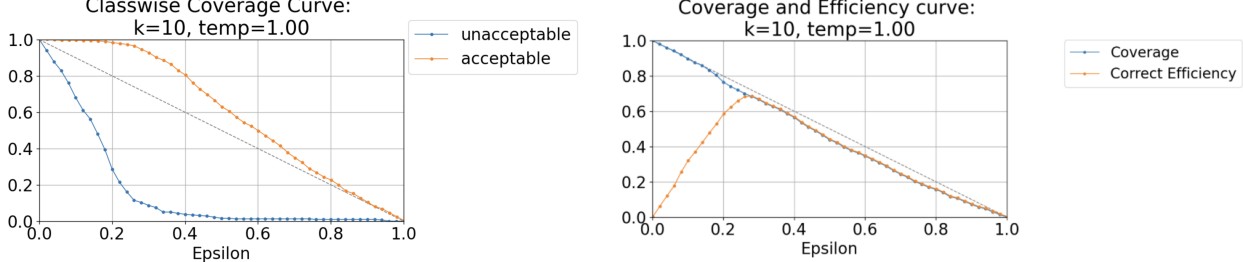

Figure 6: Coverage curves on CoLA using RoBERTa models. Undercoverage and classwise calibration perform marginally better than smaller models but still poorly.

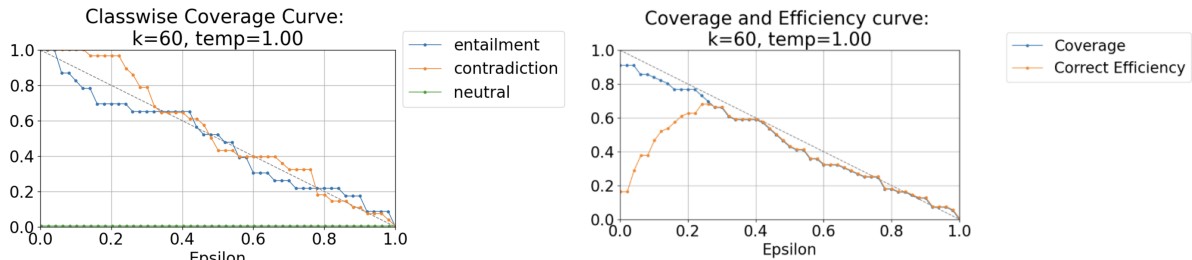

Figure 7: RoBERTa-large on CB (classwise): `neutral` class coverage fails entirely for neutral labels despite overall decent accuracy.

### 5.6 Prior Method Comparisons

We also present marginal coverage graphs for the CoLA and BoolQ datasets using prior methods. Figs. 8a–9b show their classwise coverage behavior across model and method combinations.

On CoLA, NM2 similarly exhibits significant calibration issues across both BERT-tiny and RoBERTa-base backbones. As can be seen from Fig. 8, the "unacceptable" class suffers from consistent undercoverage throughout the $\varepsilon$ range, with coverage falling far below the $1 - \varepsilon$ diagonal. This trend persists in Fig. 8b, where even the stronger RoBERTa-base model fails to improve the validity of the "unacceptable" class. These results mirror the classwise failures of CONFIDE on CoLA and indicate that the inherent ambiguity of the task and label noise challenge all distance-based methods.

On BoolQ, both NM2 and VanillaNN fail to provide reliable classwise coverage for BERT-tiny. In Fig. 9a, NM2's performance is poor for the "false" class, with coverage dropping steadily below the diagonal. The "true" class fares slightly better but still fails to meet the theoretical guarantee. VanillaNN, shown in Fig. 9b, demonstrates similarly imbalanced behavior — while both classes begin near full coverage at low $\varepsilon$, performance rapidly degrades. The "false" class undercovers more severely, likely due to the method's lack of calibrated similarity scaling.

Clearly, these results highlight how both prior conformal methods have similar classwise calibration failures on tasks with label imbalance or semantic ambiguity. Across methods, the most severe coverage violations occur for minority or hard-to-learn classes, indicating that the primary bottleneck may not lie in the conformal framework but in the quality and richness of underlying embeddings.

### 5.7 Comparing Attention vs. Flattened CONFIDE Variants

We ablate against the *flattened* variant, which vectorizes the full hidden-state matrix of the selected layer. Due to compute/VRAM costs, we only report two representative tasks (MNLI/BERT-tiny, SST-2/BERT-small) in Table 8.

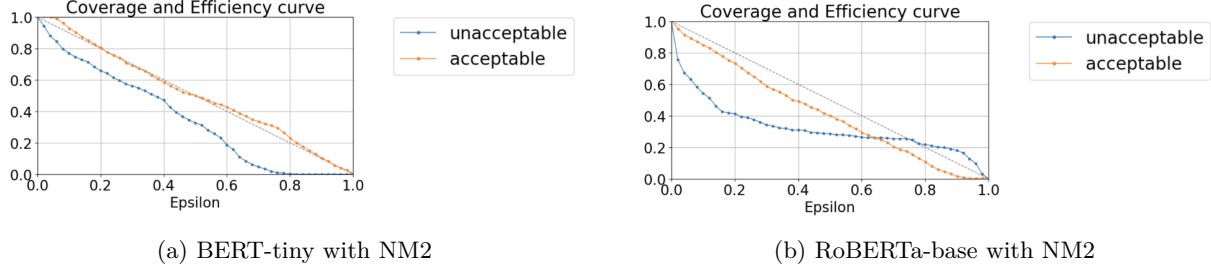

(a) BERT-tiny with NM2                (b) RoBERTa-base with NM2

Figure 8: Classwise coverage curves for CoLA using the NM2 prior method. Both BERT-tiny and RoBERTa-base show persistent undercoverage, particularly for the "unacceptable" class.

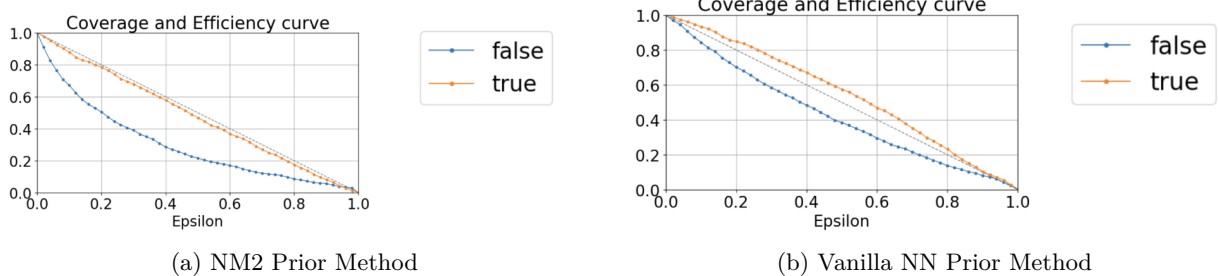

(a) NM2 Prior Method                  (b) Vanilla NN Prior Method

Figure 9: Classwise coverage curves for BoolQ using BERT-tiny with two prior methods. Both NM2 and Vanilla NN fail to maintain valid coverage, especially for the "false" class.

Flattened features often deliver *slightly higher and more stable* correct efficiency, with top-1 accuracy generally comparable to the attention variant. Gains are most visible on smaller backbones, while on larger encoders attention is typically close. Intuitively, retaining token-level structure (not only `[CLS]`) improves the separability of conforming vs. nonconforming examples and reduces sensitivity to task-specific pooling. Future research should better analyze the differences in the semantic richness of flattened vs. attention layers.

Therefore, when memory/latency budgets allow, prefer the *flattened* variant for its modest but consistent improvements in correct efficiency. Under tighter budgets, the *attention* variant offers a substantially smaller footprint with near-parity accuracy on larger encoders. On small/backbone-limited models, CONFIDE raises top-1 accuracy by up to **4.09%** (BERT-tiny) and **2.69%** (BERT-small) relative to the strongest prior baseline, with the exact hyperparameter depending on the model and dataset combination. On RoBERTa-base/large it maintains accuracy and increases correct efficiency (e.g., MRPC: $0.8627 \rightarrow 0.8848$ and $0.8578 \rightarrow 0.8652$). In general, if memory allows, we should use the **flattened** variant, which has a slightly higher and more stable correct efficiency. Under tight memory/runtime budgets, we should use the **attention** variant.

## 6 Discussions and Limitation

Despite its strengths, CONFIDE suffers from persistent calibration failures in real-world datasets and can be computationally burdensome. As can be seen from our comprehensive results, CONFIDE often suffers from significant *coverage violations*, frequently falling below the expected $(1 - \epsilon)$ threshold, especially for minority classes. These violations highlight a key limitation of CP in practice, both in past methods and in CONFIDE: The assumption of exchangeability is often not met, particularly in datasets with inherent class imbalance or input distribution drift. This limitation suggests that in safety-critical domains, conformal methods must be combined with explicit checks on calibration behavior per class or region of the input space. Otherwise, falsely assuming valid coverage can lead to underrepresented groups being frequently misclassified.

Table 8: Comparison of CONFIDE-A and CONFIDE-C using Attention and Flattened representations for MNLI (BERT-tiny) and SST-2 (BERT-small). Bolded values highlight the best result for each task and metric.

| Task / Variant | Attention | | Flattened | |
|---|---|---|---|---|
| | Acc. | Corr. Eff. | Acc. | Corr. Eff. |
| *MNLI – BERT-TINY* | | | | |
| CONFIDE-A | 0.6982 | **0.6907** | **0.6993** | 0.6497 |
| CONFIDE-C | 0.6975 | 0.6932 | **0.6980** | **0.6971** |
| *SST-2 – BERT-SMALL* | | | | |
| CONFIDE-A | 0.8933 | 0.8647 | **0.8968** | **0.8853** |
| CONFIDE-C | 0.8899 | 0.8911 | **0.8933** | **0.8933** |

Empirically, CONFIDE often performs best when probing middle transformer layers. For instance, in RoBERTa-base on CoLA (Fig. 6), the optimal performance arises at $k = 60$, $t = 1.0$, and $l = 190$, within the *encoder.layer.10* attention block, out of 12 total attention blocks. Crucially, this reflects the representational quality of that specific layer, i.e., richer class-separating embeddings, rather than a generic effect of shallow depth. In other words, performance is sensitive to the choice of intermediate layer, which introduces some instability and tuning cost. We select the layer empirically through systematic sweeps across transformer depths, with the best-performing configuration chosen based on calibration coverage and prediction set efficiency. While this procedure identifies clear trends, such as optimal layers often occurring in early-to-mid depths, it can vary slightly across random seeds or datasets. Practical mitigations include restricting the search to a few representative layers (e.g., one per transformer block), averaging nonconformity scores across adjacent layers to smooth variability, or prioritizing layers that balance accuracy with computational cost.

This trade-off between interpretability and efficiency is magnified in the `flattened` variant of CONFIDE, which extracts token-level embeddings across the full sequence length. As discussed in Section 5.7, flattened representations can outperform attention-based ones, however, they can increase memory usage and make pre-processing expensive due to their sequence-level granularity. Results are also hard to extrapolate due to the expense of storing and compare high-dimensional token-level vectors across long input sequences, running into GPU memory limits on RoBERTa models and long-context tasks such as BoolQ.

Importantly, CONFIDE assumes access to an intermediate representation $f_\ell(x)$ (hidden states or token embeddings). Most commercial LLM application programming interfaces (APIs) (e.g., GPT–4o, Gemini–2.5) do not expose these internals (and often not logits), which presents a deployment limitation rather than a methodological one. The framework is primarily designed for encoder-style, open-weights backbones where hidden states are accessible and reproducible (OpenAI, 2024; Gemini Team, 2025). Interpretability and uncertainty evaluation in open models are prerequisites for credible trust metrics in closed models. Hence, strengthening representation-level CP in open models establishes the methodological foundation required to extend these guarantees across the broader ecosystem of LLMs. In API-only contexts, CONFIDE can also remain applicable through (i) exported embeddings in place of $f_\ell(x)$ or (ii) output-only nonconformity scores (e.g., margins or likelihood gaps).

# 7 Conclusions and Future Work

This paper introduced CONFIDE, a CP framework designed to bring principled uncertainty quantification and interpretable prediction sets to transformer-based language models. Built upon the CONFINE algorithm, CONFIDE adapts these techniques to the unique architecture and representational structure of transformers, particularly BERT and RoBERTa variants. CONFIDE enables users to generate prediction sets that are transparent and explainable through nearest-neighbor retrievals.

Across a suite of GLUE and SuperGLUE tasks, CONFIDE outperforms traditional uncertainty baselines in both accuracy and correct efficiency. On most models, CONFIDE improves prediction set quality without

sacrificing top-1 accuracy. Our empirical results show that CONFIDE variants with [CLS] embeddings and class-conditional nonconformity measures outperform NM2 and VanillaNN in both aggregate and classwise coverage. These gains are especially pronounced on tasks like BoolQ and RTE. While every method suffers from undercoverage for underrepresented classes, CONFIDE attempts to make these imbalances legible through classwise coverage curves and credibility metrics, surfacing issues that remain hidden in aggregate statistics.

The broader value of conformal prediction lies in its potential to deliver trustworthy and introspective AI. In high-stakes domains like healthcare, conformal methods can output reliable differential diagnoses with guaranteed coverage, enabling practitioners to weigh multiple outcomes without over-reliance on a single prediction. One example is a model that outputs "pneumonia, bronchitis" with 95% confidence based on a series of symptoms, better than simply offering just "pneumonia" at 98% probability. CONFIDE enhances trust by providing these instance-level explanations: potentially the most similar past patients or cases in the calibration data.

Robotics offers another compelling application area. In real-world systems, such as autonomous vehicles, safety depends not only on the accuracy of decisions but also on the model's ability to know when *not* to act. CONFIDE enables safety-aware behavior by using prediction set size as a proxy for certainty: If the prediction set is large or ambiguous, the robot may default to a safer fallback action. This is important for applications like object classification in unstructured environments or decision-making under partial observability, where taking the wrong action can have irreversible consequences.

The promise of CP in language models extends even further. Future work can extend CONFIDE's ideas to generative settings (e.g., conformal decoding), multimodal tasks (e.g., image-text retrieval), and structured outputs (e.g., entity spans or logical forms), thereby expanding the reach of conformal reasoning into the core of next-generation AI systems.

In sum, CONFIDE presents robust metrics of CP methods across datasets and models, taking a step towards more interpretable, robust, and theoretically grounded NLP systems. It equips users with prediction sets they can trust and explanations they can audit without needing to modify the underlying model architecture. While computational costs remain a barrier to real-time deployment, especially at earlier layers, this tradeoff is often justified in applications where transparency and reliability are paramount. By bridging deep language models and conformal inference, CONFIDE contributes a step toward introspective, explainable, and reliable AI.

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

# Appendix

## A   Model and Dataset Details

**Benchmark selection.**   We evaluate CONFIDE on a suite of sentence- and passage-level classification tasks drawn from the GLUE and SuperGLUE benchmarks. These datasets are widely used in NLP research and present varied challenges in terms of linguistic complexity, dataset size, number of classes, and label ambiguity. All tasks selected are classification-based, ensuring compatibility with CP methods that require discrete label spaces. By testing CONFIDE on this diverse set, we validate its robustness, interpretability, and generalizability across real-world NLP scenarios.

**GLUE datasets (Wang et al., 2019)**

- **SST-2 (Stanford Sentiment Treebank v2):** Binary sentiment classification of movie reviews. Contains approximately 67,000 training examples. Well-curated and widely used for benchmarking sentence-level sentiment classification.

- **MNLI (Multi-Genre Natural Language Inference):** Three-class classification (entailment, neutral, contradiction) with over 392,000 examples. Features premise-hypothesis pairs from multiple domains (e.g., fiction, government), testing general NLI capabilities.

- **CoLA (Corpus of Linguistic Acceptability):** Binary acceptability judgment task using expert-labeled grammatical sentences. Relatively small with roughly 8,500 examples, challenging due to linguistic nuance.

- **MRPC (Microsoft Research Paraphrase Corpus):** Binary classification of whether sentence pairs are semantically equivalent. Contains around 3,700 examples and introduces noise due to label ambiguity.

- **QQP (Quora Question Pairs):** Large-scale binary paraphrase classification with over 400,000 sentence pairs. Includes noisy user-generated data, simulating web-scale text inference tasks.

- **QNLI (Question Natural Language Inference):** Binary classification derived from question-answering. Reformulated from SQuAD as sentence-pair entailment. Mid-sized with approximately 105,000 examples.

- **RTE (Recognizing Textual Entailment):** Binary entailment detection with only 2,500 examples. Known for being difficult due to label imbalance and low-resource setting.

- **WNLI (Winograd NLI):** Binary coreference reasoning task derived from the Winograd Schema Challenge. Extremely small (635 samples), difficult to model due to subtle pronoun resolution required.

**SuperGLUE Datasets (Wang et al., 2020)**

- **BoolQ (Boolean Questions):** Binary QA task requiring yes/no answers to naturally occurring queries paired with Wikipedia passages. Over 9,000 training examples, but highly noisy and linguistically ambiguous.

- **CB (CommitmentBank):** Three-class NLI task with only ~250 training examples. Requires fine-grained reasoning over implicatures in short discourse contexts. Serves as a stress test for uncertainty calibration in low-data settings.

- **MultiRC (Multi-Sentence Reading Comprehension):** Framed as a binary classification task on question-answer-context triples. Multi-label format with complex, often contradictory annotations. Approximately 9,000 QA pairs from 300 paragraphs.

# B  Hyperparameter Testing Results

## B.1  Hyperparameter Search

We performed grid search over the CONFIDE hyperparameters: *layer*, *k*, *distance metric*, *PCA*, and *temperature*. For each (model, dataset) pair, we selected the configuration that maximized either top-1 accuracy (CONFIDE-A) or correct efficiency (CONFIDE-C). Full results are visualized as heatmaps, but first, we share top-accuracy and top-correct-efficiency configurations across datasets and methods. Table 9 summarizes the key settings.

Table 9: Best CONFIDE hyperparameters across datasets and models.

| Dataset | Model | Method | Hyperparameters |
|---|---|---|---|
| glue_cola | BERT-small | CONFIDE-A | $l = 81$, $k = 10$, flattened, cosine, PCA : $No$ |
| | | CONFIDE-C | $l = 81$, $k = 10$, flattened, cosine, PCA : $No$ |
| | BERT-tiny | CONFIDE-A | $l = 16$, $k = 50$, flattened, Mahalanobis, PCA : $Yes$ |
| | | CONFIDE-C | $l = 16$, $k = 50$, flattened, Mahalanobis, PCA : $Yes$ |
| | RoBERTa-base | CONFIDE-A | $l = 190$, $k = 20$, flattened, cosine, PCA : $No$ |
| | | CONFIDE-C | $l = $ softmax, $T = 1.0$, $k = 1$, flattened, cosine, PCA : $Yes$ |
| | RoBERTa-large | CONFIDE-A | $l = 340$, $k = 1$, flattened, cosine, PCA : $No$ |
| | | CONFIDE-C | $l = 340$, $k = 1$, flattened, cosine, PCA : $No$ |
| glue_mnli | BERT-small | CONFIDE-A | $l = $ softmax, $T = 0.01$, $k = 1$, flattened, cosine, PCA : $No$ |
| | | CONFIDE-C | $l = $ softmax, $T = 0.01$, $k = 1$, flattened, cosine, PCA : $No$ |
| | BERT-tiny | CONFIDE-A | $l = $ softmax, $T = 1.0$, $k = 1$, flattened, cosine, PCA : $No$ |
| | | CONFIDE-C | $l = $ softmax, $T = 10.0$, $k = 60$, flattened, cosine, PCA : $Yes$ |
| | RoBERTa-base | CONFIDE-A | $l = 225$, $k = 10$, flattened, cosine, PCA : $No$ |
| | | CONFIDE-C | $l = 225$, $k = 1$, flattened, cosine, PCA : $No$ |
| glue_mrpc | BERT-small | CONFIDE-A | $l = 52$, $k = 20$, flattened, cosine, PCA : $No$ |
| | | CONFIDE-C | $l = 70$, $k = 20$, flattened, cosine, PCA : $No$ |
| | BERT-tiny | CONFIDE-A | $l = 34$, $k = 50$, flattened, Mahalanobis, PCA : $No$ |
| | | CONFIDE-C | $l = 16$, $k = 5$, flattened, Mahalanobis, PCA : $No$ |
| | RoBERTa-base | CONFIDE-A | $l = 189$, $k = 50$, flattened, cosine, PCA : $No$ |
| | | CONFIDE-C | $l = 191$, $k = 5$, flattened, Mahalanobis, PCA : $Yes$ |
| | RoBERTa-large | CONFIDE-A | $l = $ softmax, $T = 20.0$, $k = 1$, flattened, cosine, PCA : $No$ |
| | | CONFIDE-C | $l = $ softmax, $T = 40.0$, $k = 1$, flattened, cosine, PCA : $No$ |
| glue_qnli | BERT-small | CONFIDE-A | $l = 85$, $k = 40$, flattened, cosine, PCA : $No$ |
| | | CONFIDE-C | $l = 82$, $k = 1$, flattened, cosine, PCA : $No$ |
| | BERT-tiny | CONFIDE-A | $l = 27$, $k = 40$, flattened, cosine, PCA : $No$ |
| | | CONFIDE-C | $l = 27$, $k = 1$, flattened, cosine, PCA : $Yes$ |
| | RoBERTa-base | CONFIDE-A | $l = 225$, $k = 5$, flattened, cosine, PCA : $No$ |
| | | CONFIDE-C | $l = $ softmax, $T = 0.01$, $k = 5$, flattened, cosine, PCA : $No$ |
| glue_qqp | BERT-small | CONFIDE-A | $l = 81$, $k = 1$, flattened, cosine, PCA : $No$ |
| | | CONFIDE-C | $l = $ softmax, $T = 0.01$, $k = 40$, flattened, cosine, PCA : $No$ |
| | BERT-tiny | CONFIDE-A | $l = 27$, $k = 1$, flattened, cosine, PCA : $No$ |
| | | CONFIDE-C | $l = 27$, $k = 1$, flattened, cosine, PCA : $No$ |
| | RoBERTa-base | CONFIDE-A | $l = 225$, $k = 20$, flattened, cosine, PCA : $No$ |
| | | CONFIDE-C | $l = 225$, $k = 1$, flattened, cosine, PCA : $No$ |
| glue_rte | BERT-small | CONFIDE-A | $l = 38$, $k = 60$, flattened, Mahalanobis, PCA : $Yes$ |
| | | CONFIDE-C | $l = 38$, $k = 60$, flattened, Mahalanobis, PCA : $Yes$ |
| | BERT-tiny | CONFIDE-A | $l = 27$, $k = 50$, flattened, cosine, PCA : $Yes$ |
| | | CONFIDE-C | $l = 38$, $k = 50$, flattened, cosine, PCA : $Yes$ |
| | RoBERTa-base | CONFIDE-A | $l = 190$, $k = 1$, flattened, cosine, PCA : $No$ |
| | | CONFIDE-C | $l = 189$, $k = 40$, flattened, cosine, PCA : $Yes$ |
| | RoBERTa-large | CONFIDE-A | $l = 340$, $k = 20$, flattened, Mahalanobis, PCA : $Yes$ |
| | | CONFIDE-C | $l = 340$, $k = 5$, flattened, Mahalanobis, PCA : $Yes$ |
| glue_sst2 | BERT-small | CONFIDE-A | $l = 82$, $k = 1$, flattened, cosine, PCA : $Yes$ |
| | | CONFIDE-C | $l = $ softmax, $T = 10.0$, $k = 1$, flattened, cosine, PCA : $Yes$ |
| | BERT-tiny | CONFIDE-A | $l = 27$, $k = 5$, flattened, Mahalanobis, PCA : $No$ |
| | | CONFIDE-C | $l = 34$, $k = 50$, flattened, cosine, PCA : $Yes$ |
| | RoBERTa-base | CONFIDE-A | $l = $ softmax, $T = 10.0$, $k = 1$, flattened, cosine, PCA : $No$ |
| | | CONFIDE-C | $l = $ softmax, $T = 0.01$, $k = 1$, flattened, cosine, PCA : $Yes$ |
| glue_wnli | BERT-small | CONFIDE-A | $l = 16$, $k = 60$, flattened, cosine, PCA : $Yes$ |
| | | CONFIDE-C | $l = 16$, $k = 50$, flattened, cosine, PCA : $Yes$ |
| | BERT-tiny | CONFIDE-A | $l = $ softmax, $T = 0.01$, $k = 1$, flattened, cosine, PCA : $No$ |
| | | CONFIDE-C | $l = $ softmax, $T = 0.01$, $k = 1$, flattened, cosine, PCA : $No$ |
| | RoBERTa-large | CONFIDE-A | $l = 45$, $k = 1$, flattened, Mahalanobis, PCA : $Yes$ |
| | | CONFIDE-C | $l = 45$, $k = 1$, flattened, cosine, PCA : $No$ |
| superglue_boolq | BERT-small | CONFIDE-A | $l = $ softmax, $T = 10.0$, $k = 1$, flattened, cosine, PCA : $No$ |
| | | CONFIDE-C | $l = $ softmax, $T = 0.01$, $k = 5$, flattened, cosine, PCA : $No$ |
| | BERT-tiny | CONFIDE-A | $l = 16$, $k = 5$, flattened, Mahalanobis, PCA : $No$ |
| | | CONFIDE-C | $l = 16$, $k = 5$, flattened, Mahalanobis, PCA : $No$ |
| | RoBERTa-base | CONFIDE-A | $l = $ softmax, $T = 0.01$, $k = 1$, flattened, cosine, PCA : $Yes$ |
| | | CONFIDE-C | $l = $ softmax, $T = 0.01$, $k = 1$, flattened, cosine, PCA : $Yes$ |
| superglue_cb | BERT-small | CONFIDE-A | $l = 9$, $k = 5$, flattened, cosine, PCA : $Yes$ |

*Continued on next page*

| Dataset | Model | Method | Hyperparameters |
|---------|-------|--------|-----------------|
| superglue_multirc | BERT-tiny | CONFIDE-C | $l = 27$, $k = 1$, flattened, cosine, PCA : *No* |
| | | CONFIDE-A | $l = 16$, $k = 5$, flattened, cosine, PCA : *Yes* |
| | RoBERTa-base | CONFIDE-C | $l = 16$, $k = 10$, flattened, cosine, PCA : *Yes* |
| | | CONFIDE-A | $l = 225$, $k = 1$, flattened, Mahalanobis, PCA : *No* |
| | RoBERTa-large | CONFIDE-C | $l = 225$, $k = 1$, flattened, Mahalanobis, PCA : *No* |
| | | CONFIDE-A | $l = 106$, $k = 60$, flattened, cosine, PCA : *No* |
| | BERT-small | CONFIDE-C | $l = 106$, $k = 60$, flattened, cosine, PCA : *No* |
| | | CONFIDE-A | $l = 85$, $k = 50$, flattened, cosine, PCA : *No* |
| | BERT-tiny | CONFIDE-C | $l = \text{softmax}$, $T = 0.1$, $k = 40$, flattened, cosine, PCA : *No* |
| | | CONFIDE-A | $l = 27$, $k = 5$, flattened, cosine, PCA : *No* |
| | RoBERTa-base | CONFIDE-C | $l = 38$, $k = 5$, flattened, cosine, PCA : *Yes* |
| | | CONFIDE-A | $l = 45$, $k = 60$, flattened, cosine, PCA : *No* |
| | | CONFIDE-C | $l = 45$, $k = 60$, flattened, cosine, PCA : *No* |

## B.2 Layer Index to Canonical Name

In this section, we map layer index numbers to their semantic alias.

**BERT-tiny (BoolQ)**

| $l$ | Semantic alias |
|-----|----------------|
| 16 | block.0.attn_out |
| 27 | block.1 |
| 34 | block.1.attn_out |
| 38 | block.1.mlp_in |
| 41 | block.1.output |
| – | softmax |

**BERT-small (BoolQ)**

| $l$ | Semantic alias |
|-----|----------------|
| 16 | block.0.attn_out |
| 34 | block.1.attn_out |
| 38 | block.1.mlp_in |
| 52 | block.2.attn_out |
| 70 | block.3.attn_out |
| 81 | pooler |
| 85 | classifier |
| – | softmax |

**RoBERTa-base (BoolQ)**

| $l$ | Semantic alias |
|-----|----------------|
| 45 | block.2.attn_out |
| 63 | block.3.attn_out |
| 65 | block.3.mlp_in |
| 99 | block.5.attn_out |
| 191 | block.10.attn_in |
| 225 | classifier.out_proj |
| – | softmax |

**RoBERTa-large (BoolQ)**

| $l$ | Semantic alias |
|-----|----------------|
| 38 | block.1.mlp_in |
| 41 | block.1.output |
| 106 | block.5.attn_out |
| 340 | block.18.attn_out |
| 423 | classifier.dense |
| 424 | classifier.out_proj |
| – | softmax |

## B.3 Heatmaps

We include in Figs. 10-15 a subset of experimental plots organized by dataset and model. For each grid parameter of $k$ and $l$, we present the best accuracy and best top correct efficiency, found at *any* combination of the other hyperparameters, including flattened vs. attention, cosine vs. Mahalanobis, or PCA true vs. false. Similarly, for softmax layers, we do the same for $k$ vs. $T$.

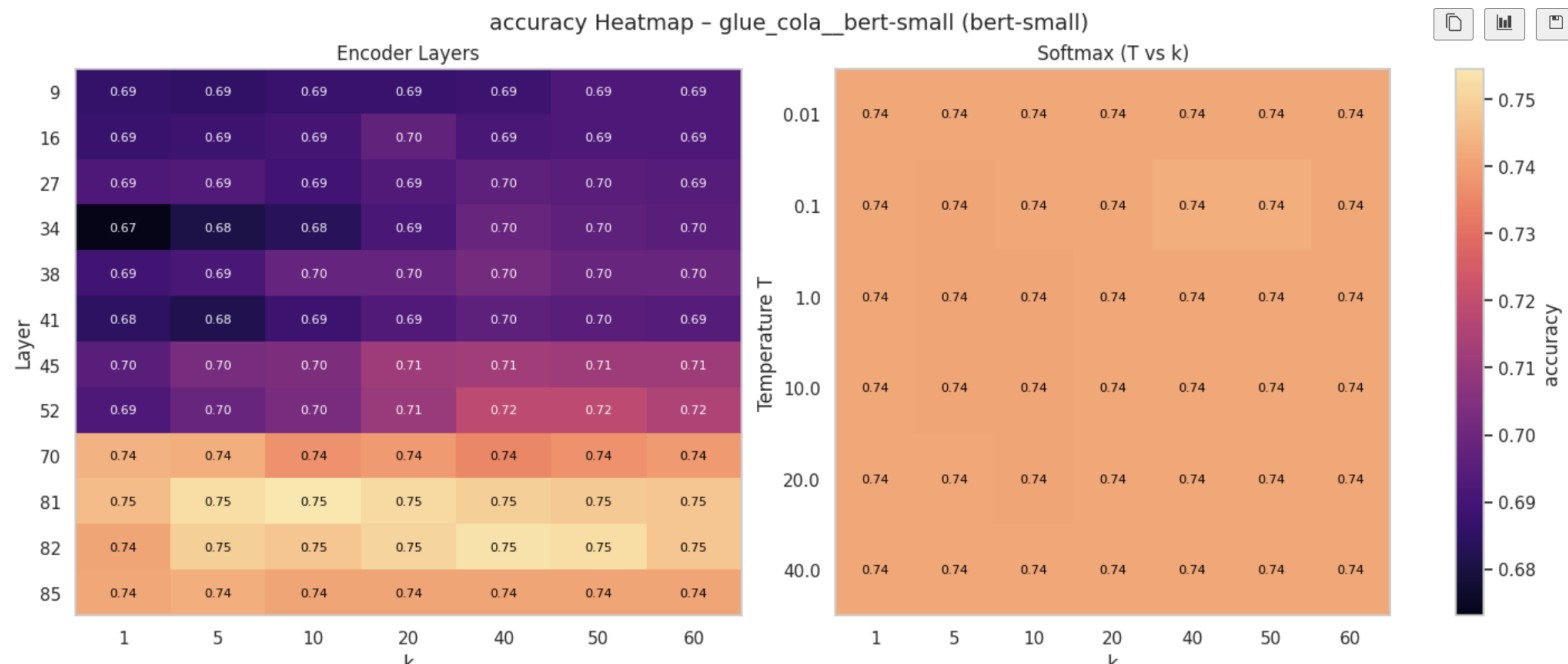

Figure 10: BERT-small hyperparameter testing on the CoLA dataset - accuracy

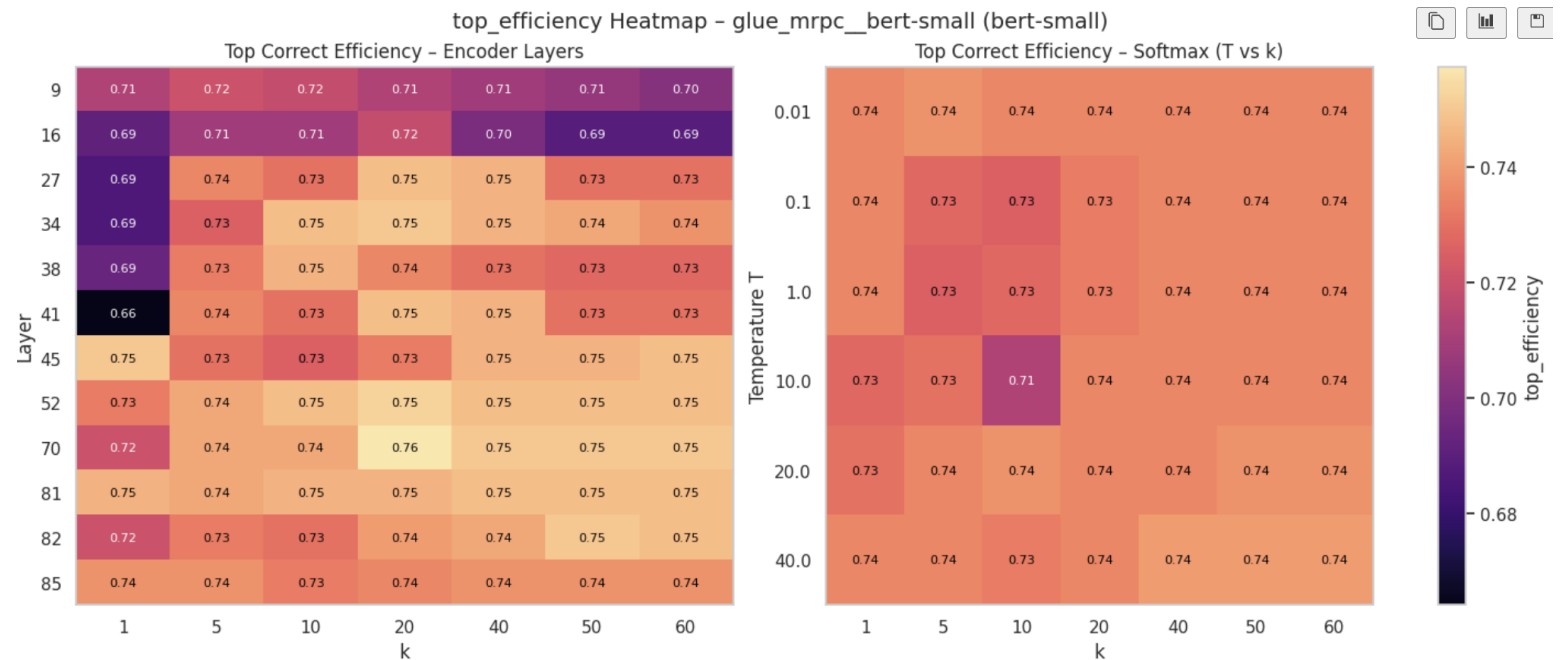

Figure 11: BERT-small hyperparameter testing on the MRPC dataset - correct efficiency

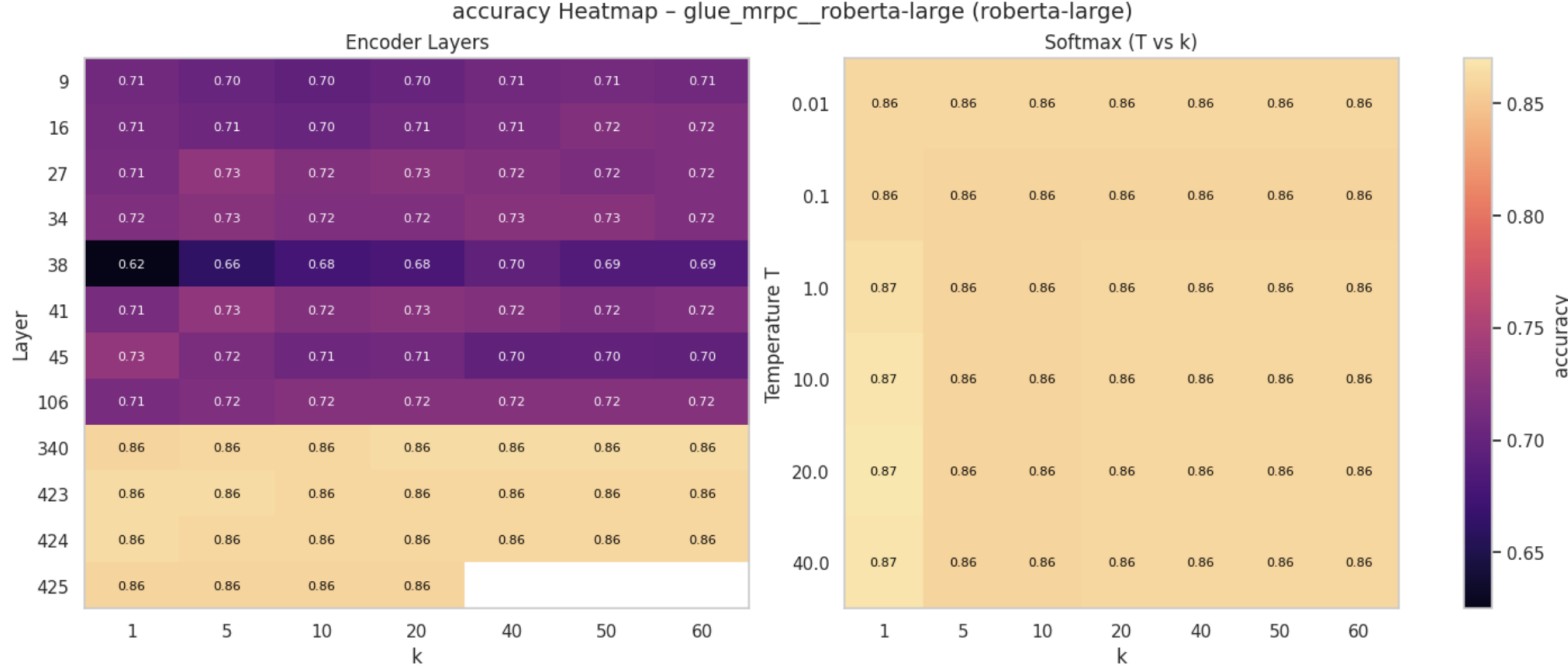

Figure 12: RoBERTa-large hyperparameter testing on the MRPC dataset - accuracy

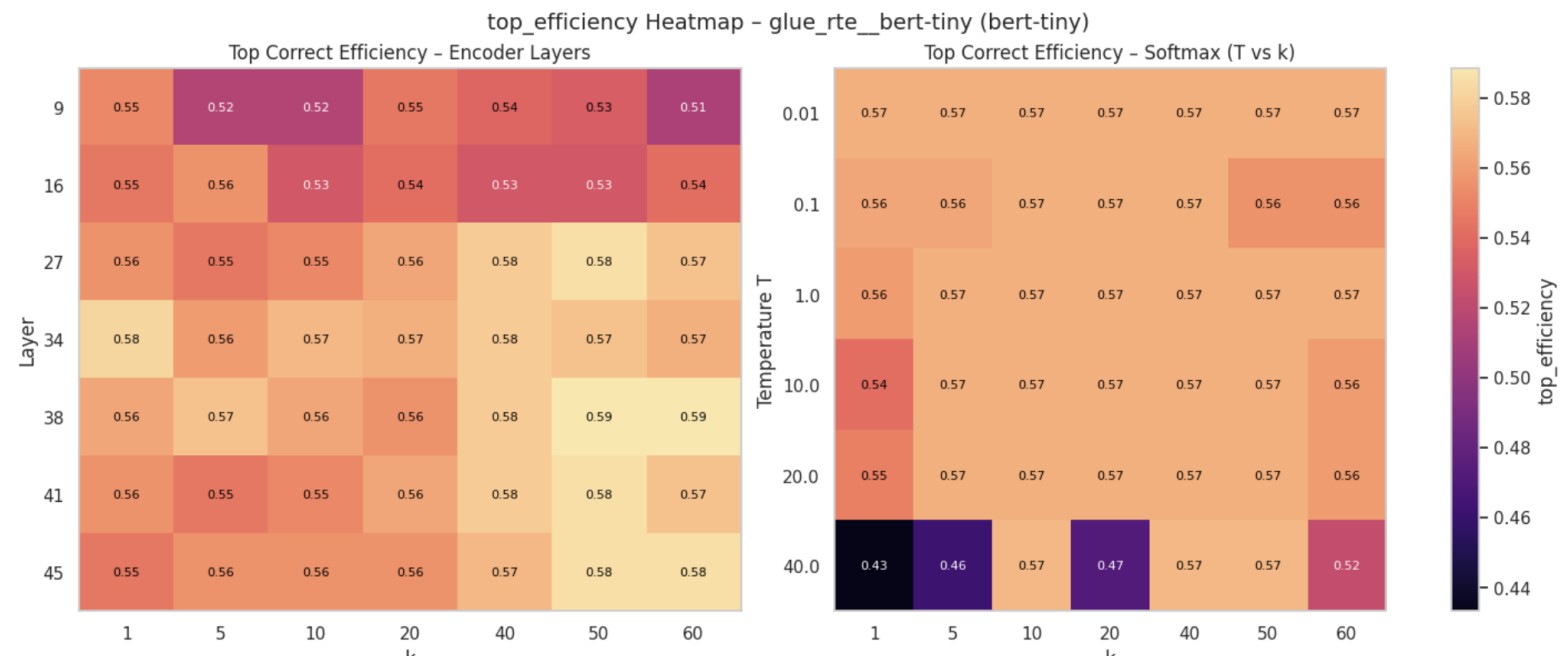

Figure 13: BERT-tiny hyperparameter testing on the RTE dataset - correct efficiency

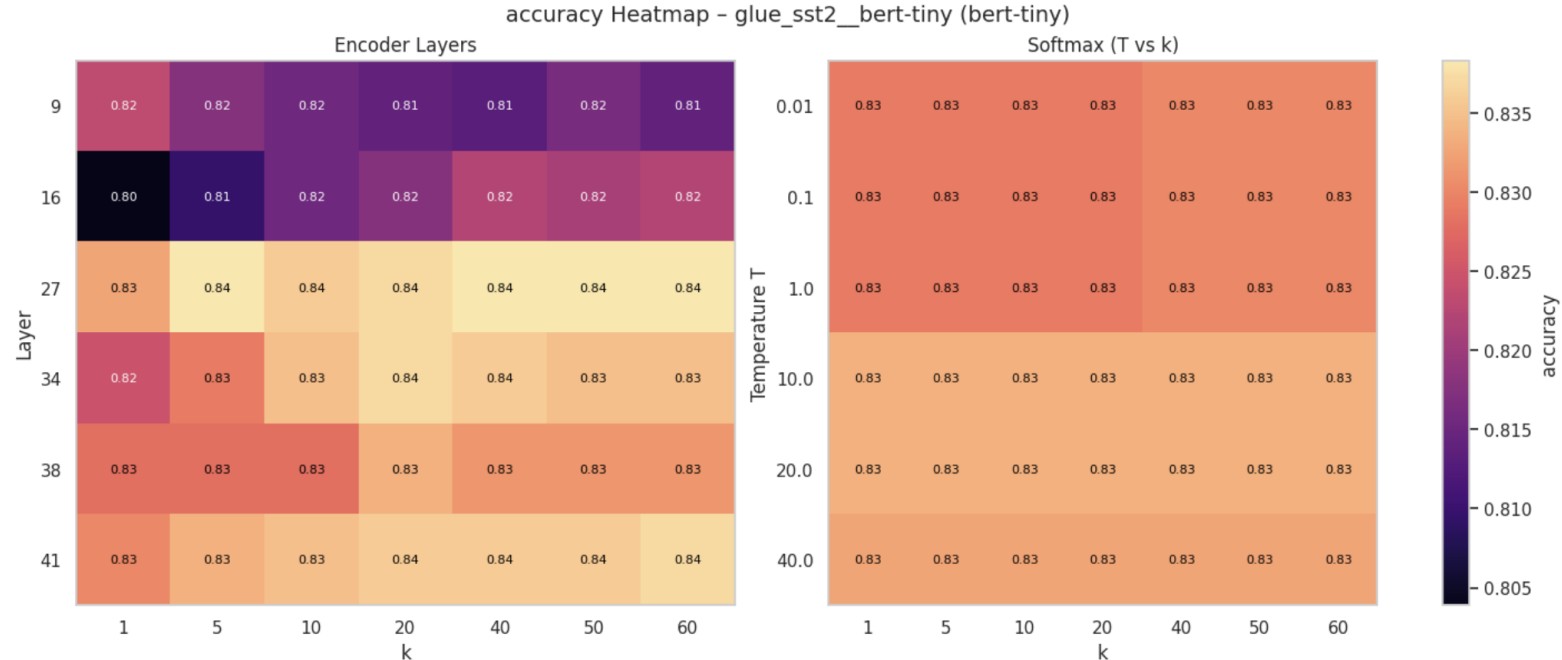

Figure 14: BERT-tiny hyperparameter testing on the SST2 dataset - accuracy

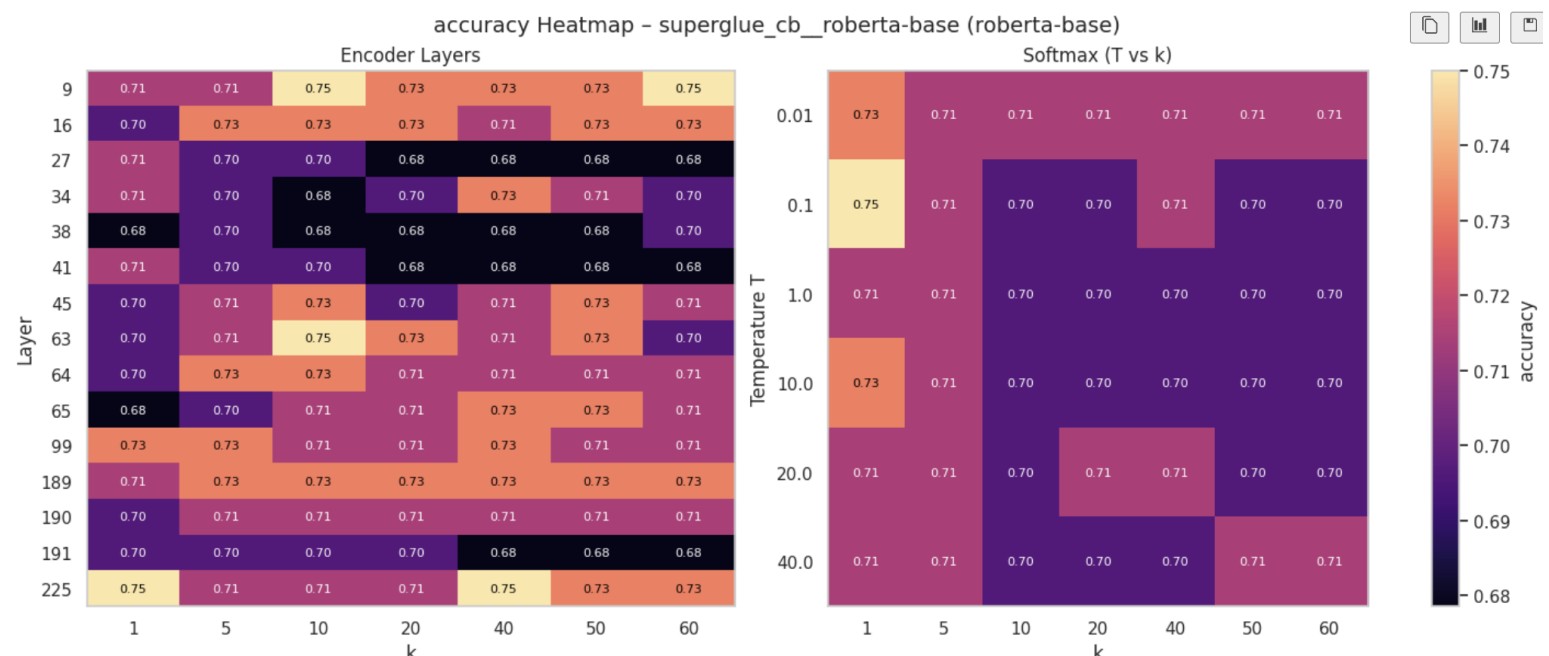

Figure 15: RoBERTa-base hyperparameter testing on the CB dataset - accuracy

## C  Timing and Computational Cost

### C.1  Computational Complexity Analysis

Our computational complexity analysis is fully based on the original CONFINE framework (Huang et al., 2025). The pre-processing phase requires every instance in the training and calibration sets to undergo a forward pass through the neural network for feature extraction. Let $d$ denote the dimensionality of the extracted feature vector, $N_t$ the size of the proper training set, and $N_c$ the size of the calibration set. If $T_f$ is the time taken for a single forward pass, then the total pre-processing time is given by $O((N_t + N_c)T_f + N_c d N_t)$. This includes both network inference and class-conditional nearest-neighbor index construction. Once feature vectors are extracted, they are stored in $O(dN_t)$ space. During inference, evaluating CONFINE on a single test point requires one additional forward pass and one nearest-neighbor lookup. This results in a total runtime of $O(T_f + dN_t)$ per test example. Compared to standard neural inference, which only incurs $O(T_f)$ time, CONFINE adds a cost of $O(dN_t)$ to enable interpretability.

### C.2  Timing Implications of Layer Choice

We measure wall-clock time on a single GPU using the same seeds and configurations as the main experiments. For each run, we log the backbone, the probed encoder block $\ell$, the sizes of the train/calibration/test splits, and the run times. Across 36 runs, calibration time scales linearly with the size of the training and calibration sets (Pearson's correlation $r \approx 0.987$), and test time scales linearly with the size of the test set (Pearson's correlation $r \approx 0.992$), indicating a high positive correlation between dataset size and run time. This aligns with our complexity derivation (Appendix C.1): per test example, CONFINE requires one forward pass plus a nearest-neighbor sweep over the reference embeddings, i.e., $O(T_f + dN_t)$. Hence, total test time grows as a function of $(T_f + dN_t)$, and since $T_f$ increases with depth, deeper probes (larger $\ell$) are slower. We track this relationship in RoBERTa-base, where a least-squares regression of time versus encoder-block index yield the slopes in Table 10. Trends are monotonic on SST-2 and QNLI; they are nearly flat on MRPC, where the small training set limits variation in nearest-neighbor search time across layers. We also compare per-example latency across model sizes on MRPC (Table 11); as expected, RoBERTa-base has higher average latency than BERT-tiny.

| Dataset | $\Delta$ calibration time / block (s) | | $\Delta$ test time / block (s) | |
|---------|------|------|------|------|
| SST-2 | 1.58 | ($R^2$=0.983) | 0.075 | ($R^2$=0.966) |
| QNLI | 5.40 | ($R^2$=0.961) | 0.980 | ($R^2$=0.994) |
| MRPC | 0.048 | ($R^2$=0.735) | 0.021 | ($R^2$=0.709) |

Table 10: RoBERTa-base timing vs depth. Slope from least-squares fit of time to encoder block index.

| Model | Dataset | Test latency (ms / example) |
|-------|---------|------|
| BERT-tiny | MRPC | 1.48 |
| RoBERTa-base | MRPC | 3.10 |

Table 11: Per-example test time averaged across probed blocks; exact $k$NN; same hardware.

## D  Additional Results

First, Fig. 16 represents a successful case of CONFINE producing a small set with high confidence. This contrasts to the example presented in Figure 1, and demonstrates how a similar semantic sentence structure yields closer "same-class" nearest neighbors, yielding a tighter prediction set.

In Tables. 12-14, we share additional performance metrics across models and datasets, many of which also result in higher accuracy and correct efficiency than prior methods.

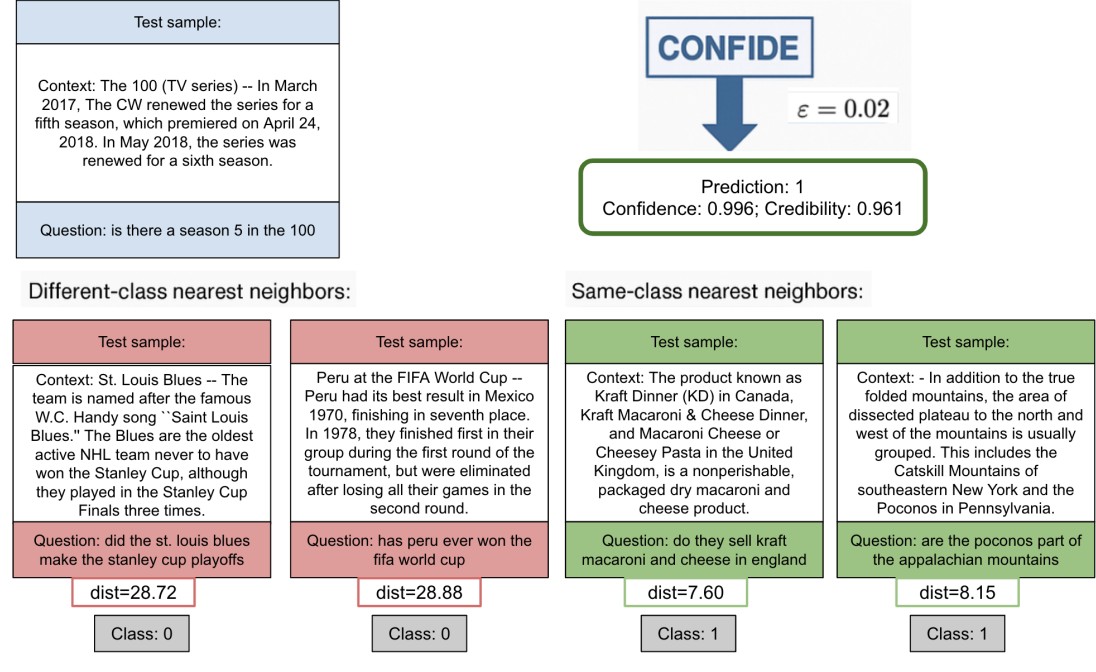

Figure 16: Success case on BoolQ (BERT-tiny). CONFIDE separates neighbors by class, yielding a confident and credible singleton prediction.

Table 12: Performance on SuperGLUE benchmarks using BERT-tiny: BoolQ, CB, and MultiRC.

| Method | BoolQ | | CB | | MultiRC | |
|---|---|---|---|---|---|---|
| | Test Acc | Top Corr Eff | Test Acc | Top Corr Eff | Test Acc | Top Corr Eff |
| Original NN | 0.6596 | – | 0.5357 | – | 0.6219 | – |
| 1-Nearest Neighbor | 0.5602 | 0.5590 | 0.4286 | 0.4286 | 0.5192 | 0.4792 |
| NM (2) | 0.6596 | 0.6529 | 0.5357 | 0.5000 | 0.6219 | 0.6209 |
| CONFIDE-A (ours) | **0.6636** | **0.6630** | **0.7321** | **0.7143** | **0.6275** | 0.5918 |
| CONFIDE-C (ours) | **0.6636** | **0.6630** | **0.7321** | **0.7143** | 0.6244 | **0.6229** |

In Figs. 17-42, we present between 1-3 model configurations, both with and without classwise split, of each dataset. Graphs are sourced from the best hyperparameter combination, found in Section B.1. Boolq results are not repeated as both best parameter combinations are already presented in Section 5.

We observe a strong match between coverage and correct efficiency across most $\epsilon$ values in the QNLI task using BERT-tiny, especially when classwise calibrated. This suggests that even with a limited model capacity, the learned representations in QNLI provide sufficient semantic separation between the "entailment" and "not-entailment" classes. QNLI tends to involve syntactically structured queries with informative lexical anchors. This consistency likely helps CONFIDE identify meaningful neighborhoods in the embedding space.

Then, despite WNLI being a notoriously noisy dataset with adversarial structure, we see surprisingly effective calibration here. The slight smoothness and separation in classwise curves imply that the selected layer (and hyperparameters) successfully isolate decision boundaries in a way that suppresses overfitting to artifacts.

Meanwhile, CONFIDE struggles with MultiRC in generating meaningful classwise separation or maintaining correct efficiency. Coverage drops off rapidly while correct efficiency remains flat, indicating that the internal representations fail to capture the task's complexity.

Table 13: Performance on GLUE benchmarks using BERT-small: CoLA, MNLI, MRPC, and QNLI.

| Method | CoLA | | MNLI | | MRPC | | QNLI | |
|---|---|---|---|---|---|---|---|---|
| | Test Acc | Top Corr Eff | Test Acc | Top Corr Eff | Test Acc | Top Corr Eff | Test Acc | Top Corr Eff |
| Original NN | 0.7421 | – | 0.7914 | 0.7805 | 0.7230 | – | 0.8691 | – |
| 1-Nearest Neighbor | 0.5791 | 0.5762 | 0.3403 | 0.3254 | 0.6103 | 0.6103 | 0.5067 | 0.5050 |
| NM1/NM2 | 0.7421 | 0.7392 | **0.7914** | 0.7805 | 0.7230 | 0.7034 | 0.8691 | 0.8671 |
| CONFIDE-A (ours) | **0.7546** | **0.7546** | 0.7904 | **0.7865** | **0.7672** | 0.7525 | **0.8704** | 0.8669 |
| CONFIDE-C (ours) | **0.7546** | **0.7546** | 0.7904 | **0.7865** | 0.7598 | **0.7574** | 0.8699 | **0.8697** |

Table 14: Performance on GLUE benchmarks using BERT-small: QQP, RTE, SST-2, and WNLI.

| Method | QQP | | RTE | | SST-2 | | WNLI | |
|---|---|---|---|---|---|---|---|---|
| | Test Acc | Top Corr Eff | Test Acc | Top Corr Eff | Test Acc | Top Corr Eff | Test Acc | Top Corr Eff |
| Original NN | 0.8893 | – | 0.5884 | – | 0.8911 | – | 0.4648 | – |
| 1-Nearest Neighbor | 0.6423 | 0.6391 | 0.4693 | 0.4549 | 0.5011 | 0.4794 | 0.2394 | 0.2394 |
| NM1/NM2 | 0.8893 | 0.8739 | 0.5884 | 0.5884 | 0.8911 | 0.8830 | 0.4648 | 0.4648 |
| CONFIDE-A (ours) | **0.8933** | 0.8820 | **0.6245** | **0.6173** | **0.8933** | 0.8647 | **0.5915** | 0.5493 |
| CONFIDE-C (ours) | 0.8901 | **0.8846** | **0.6245** | **0.6173** | 0.8899 | **0.8911** | 0.5634 | **0.5634** |

Ultimately, these examples illustrate that CONFIDE is highly adaptable across datasets and transformer architectures. However, its success is dependent on internal factors such as the semantic coherence of class labels, the quality of intermediate layer representations, and dataset-specific noise or adversarial traits. Careful calibration and model-aware design choices are critical to unlocking CONFIDE's full potential.

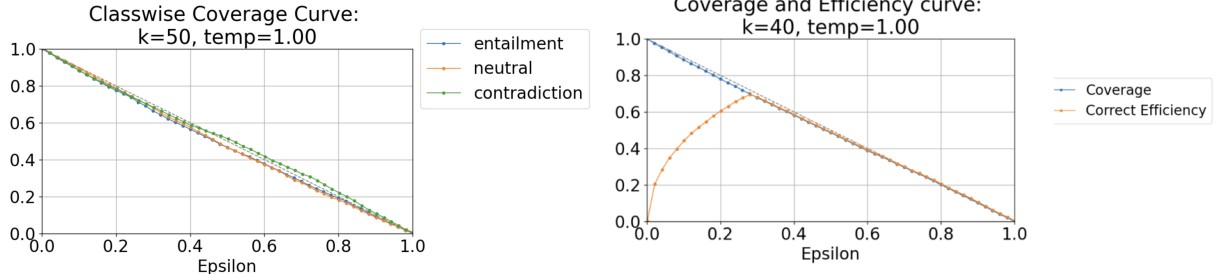

Figure 17: Classwise and aggregate coverage curves for BERT-tiny on MNLI. All classes are well calibrated and closely follow the diagonal, indicating robust conformal validity.

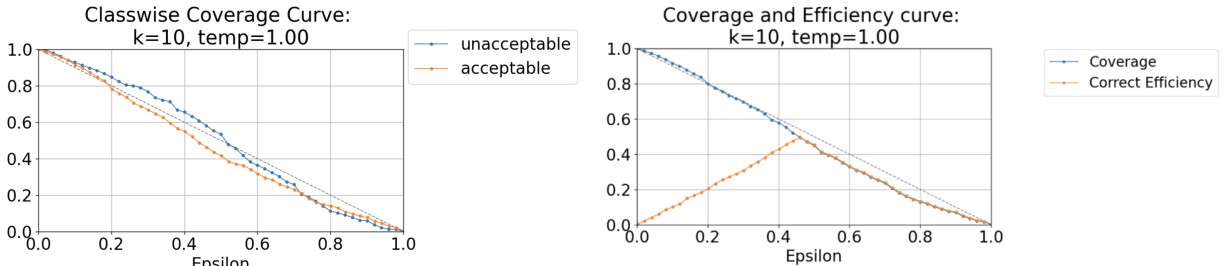

Figure 18: Improvement performances for CoLA with classwise calibration.

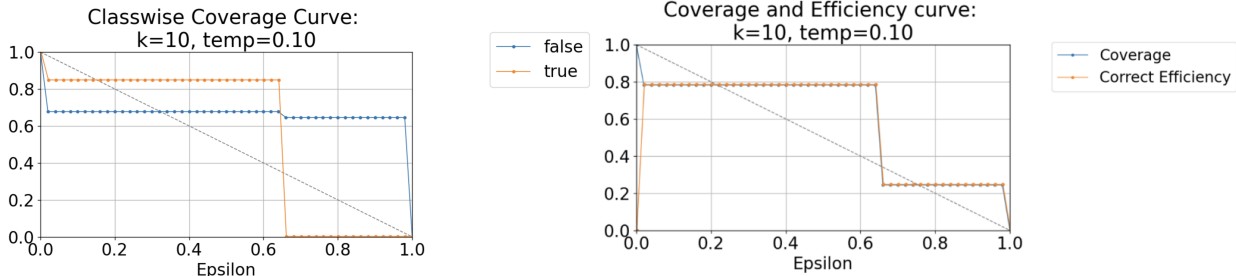

Figure 19: Improvement performances for BoolQ compared to smaller models

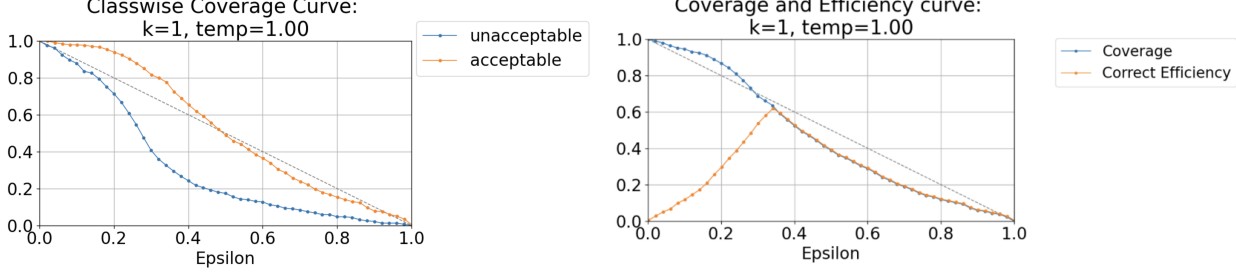

Figure 20: Coverage curves on CoLA using BERT-small models. Undercoverage and classwise calibration perform as poorly as BERT-tiny models.

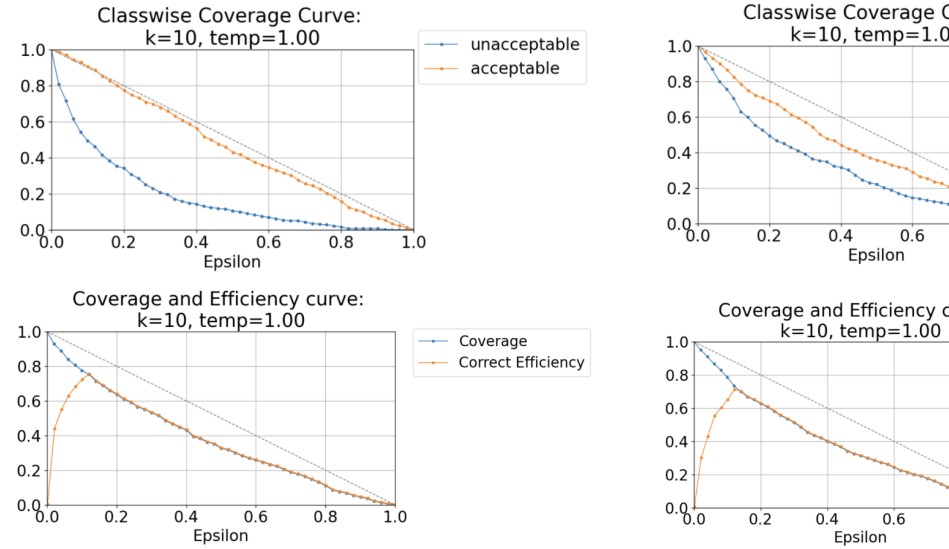

Figure 21: BERT-small CoLA classwise-false

Figure 22: BERT-small CoLA classwise-true

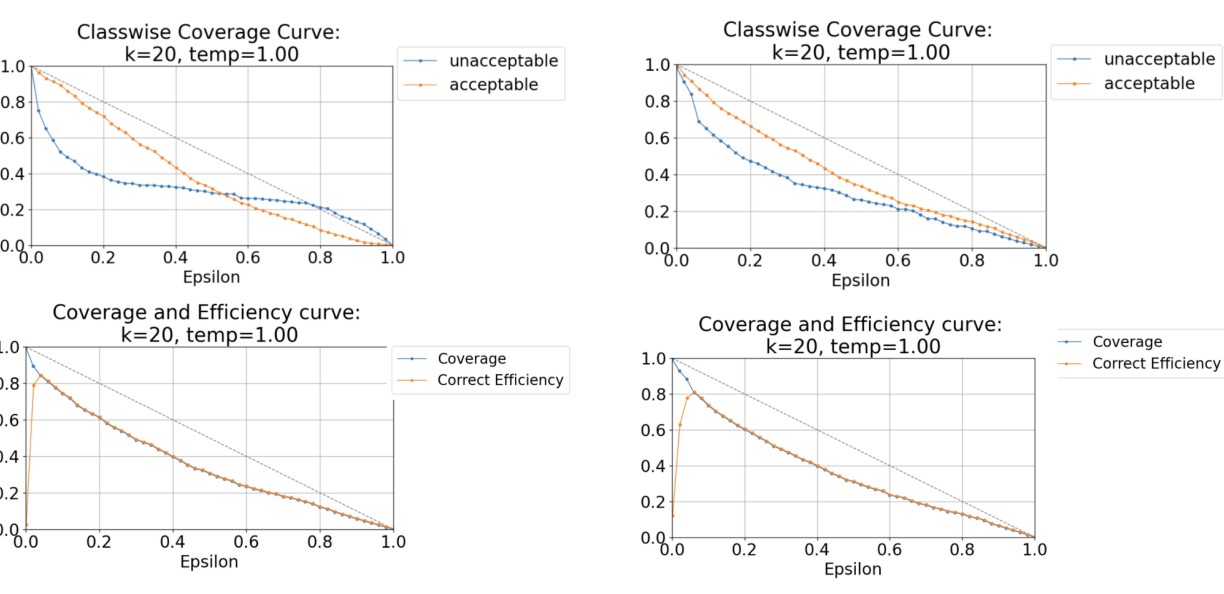

Figure 23: RoBERTa-base CoLA classwise-false

Figure 24: RoBERTa-base CoLA classwise-true

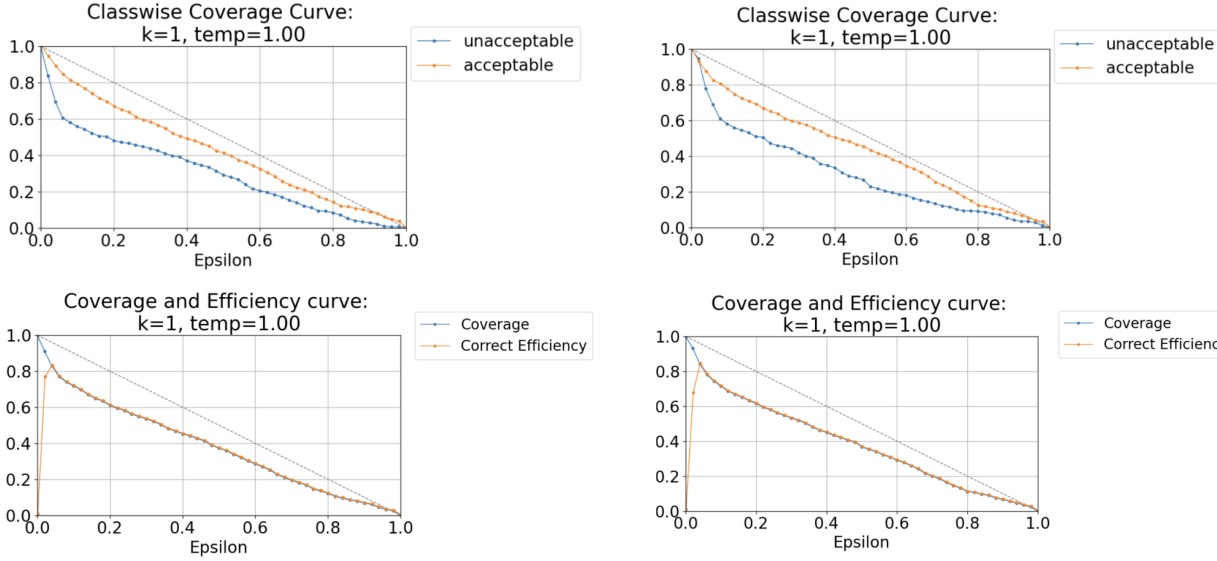

Figure 25: RoBERTa-large CoLA classwise-false

Figure 26: RoBERTa-large CoLA classwise-true

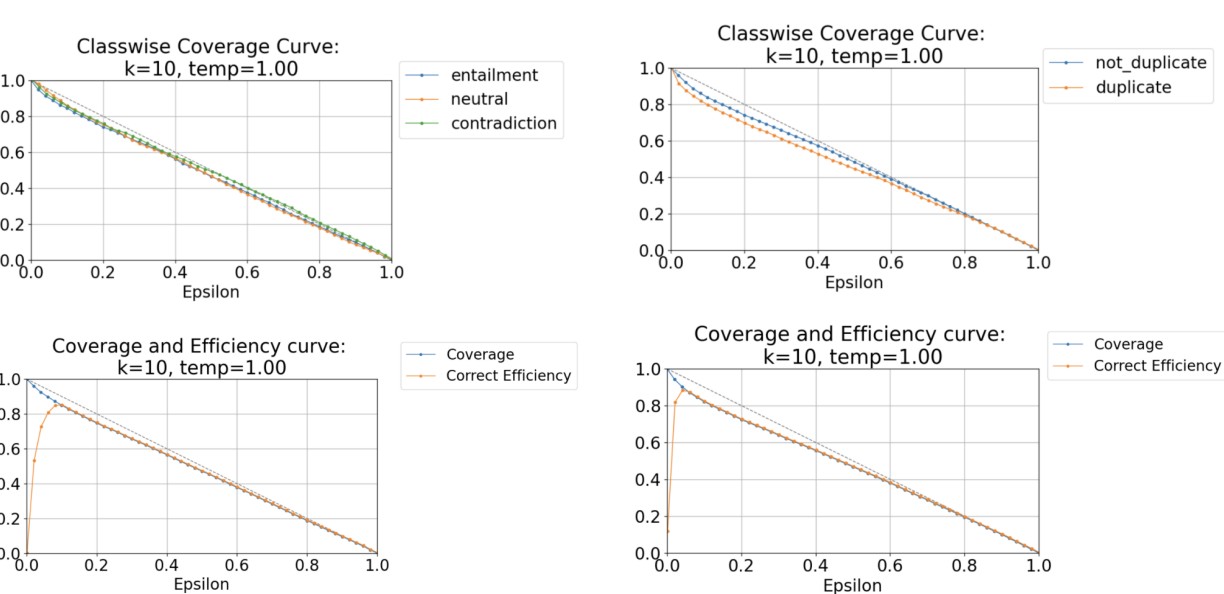

Figure 27: RoBERTa-base MNLI classwise-true

Figure 28: BERT-small QQP classwise-true

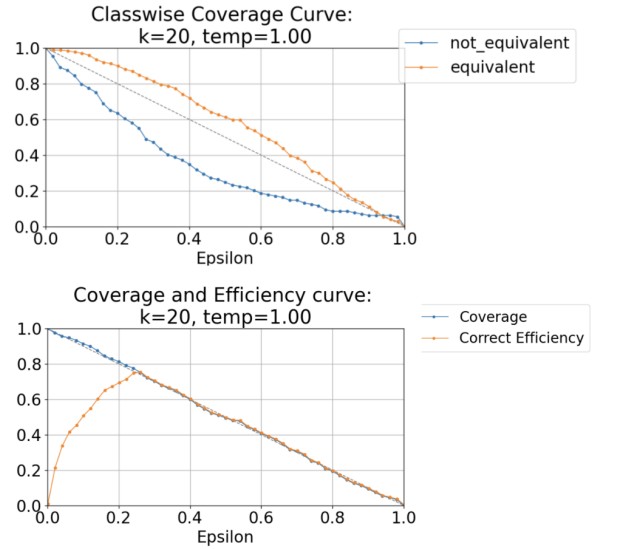

Figure 29: BERT-small MRPC classwise-false

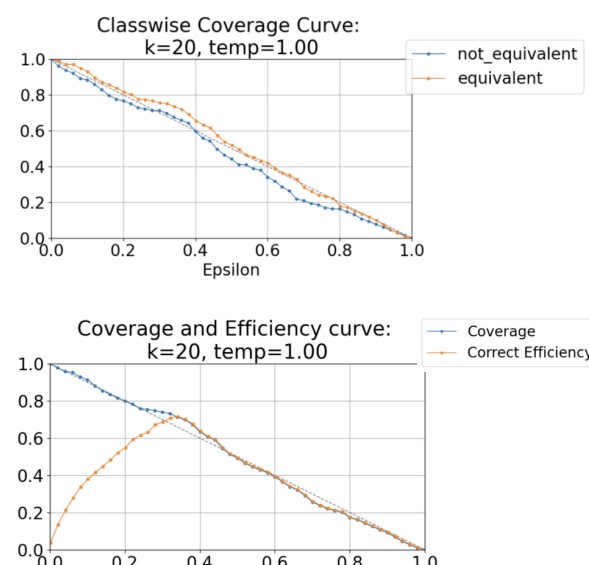

Figure 30: BERT-small MRPC classwise-true

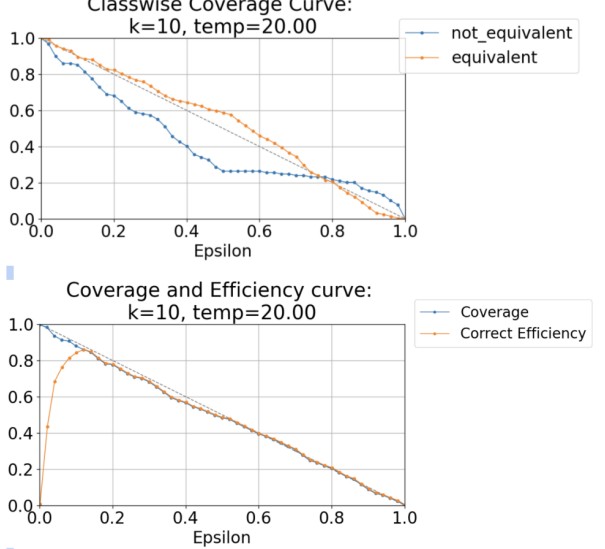

Figure 31: RoBERTa-large MRPC classwise-false

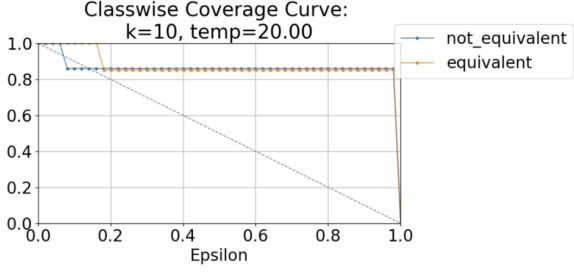

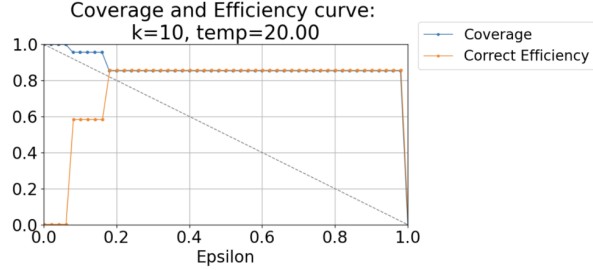

Figure 32: RoBERTa-large MRPC classwise-true

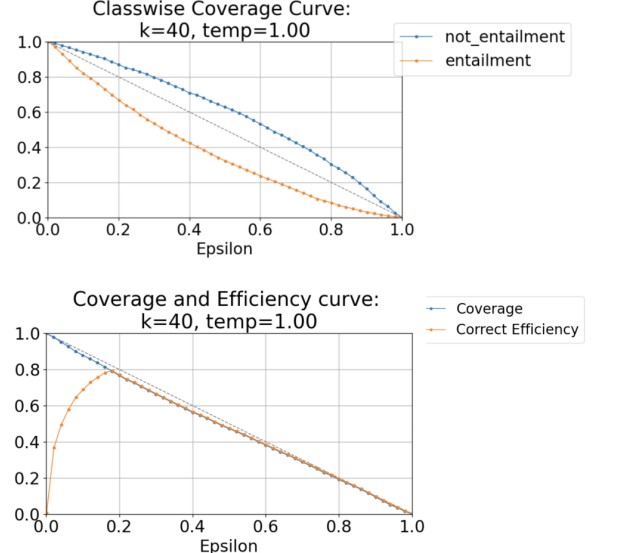
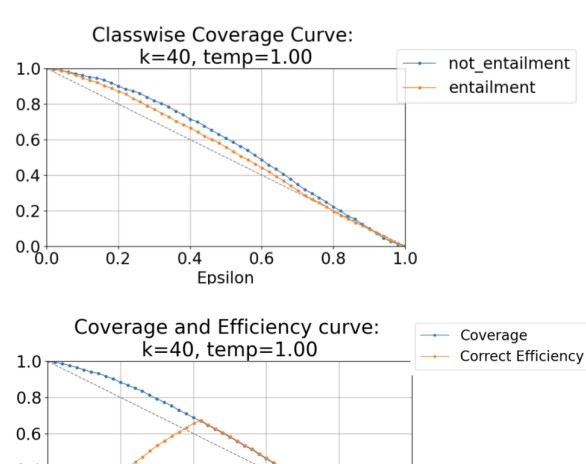
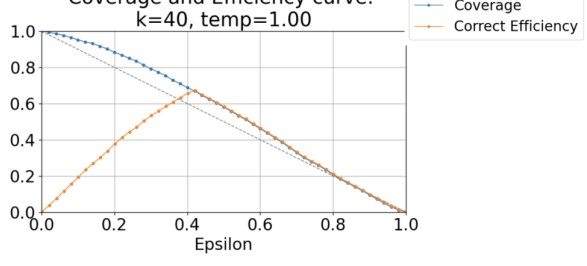

Figure 33: BERT-tiny QNLI classwise-false

Figure 34: BERT-tiny QNLI classwise-true

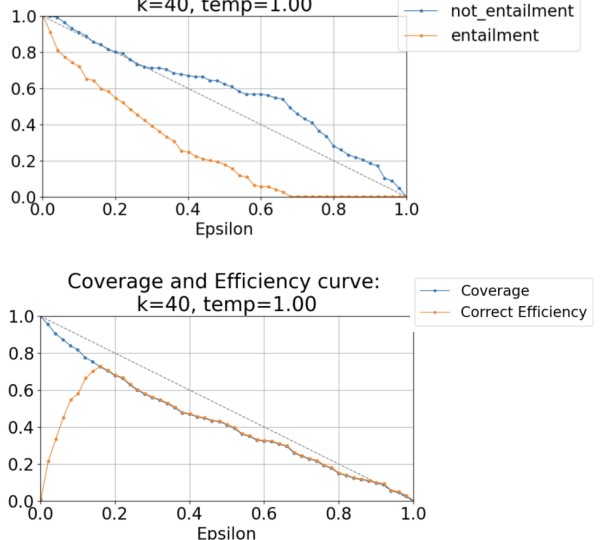
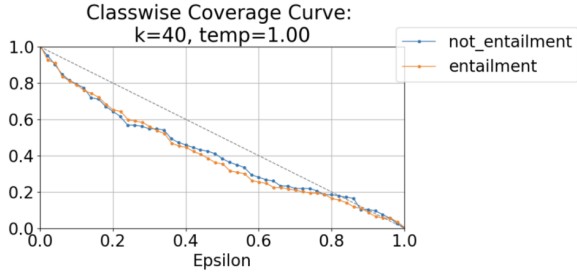
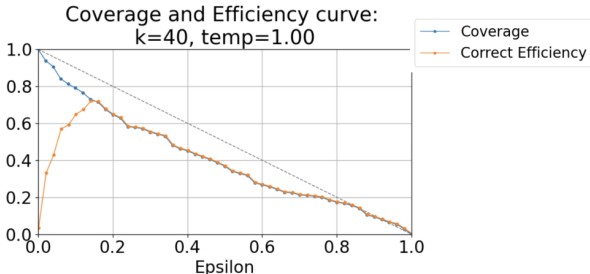

Figure 35: RoBERTa-base RTE classwise-false

Figure 36: RoBERTa-base RTE classwise-true

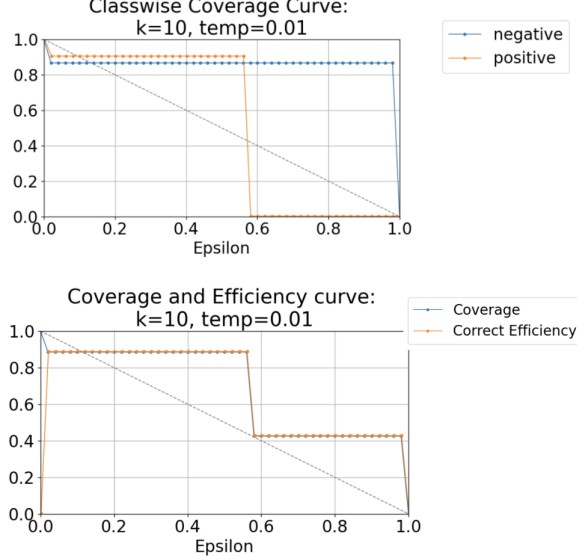

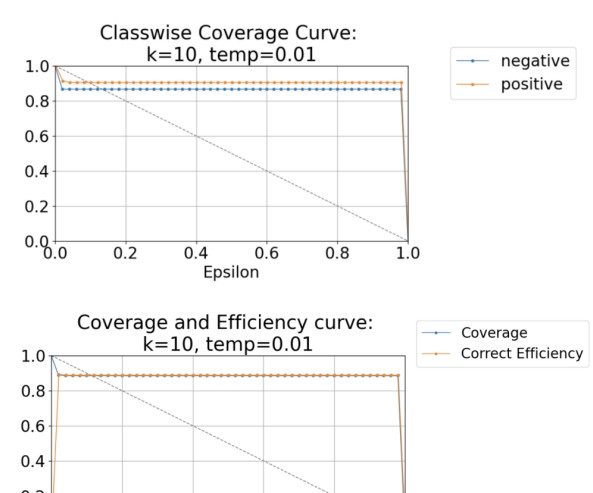

Figure 37: BERT-small SST2 classwise-false

Figure 38: BERT-small SST2 classwise-true

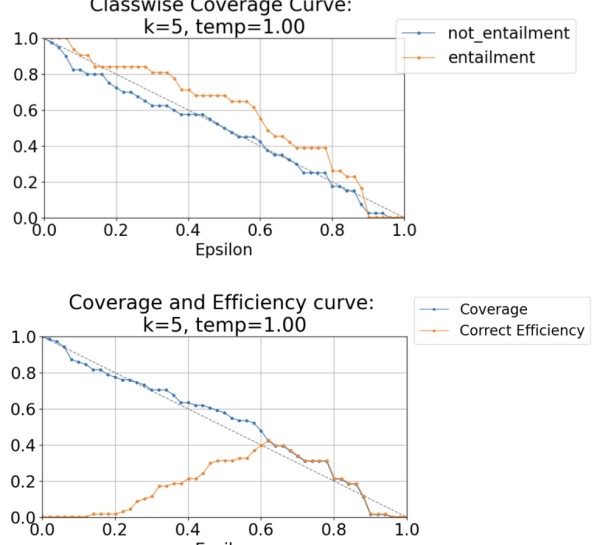

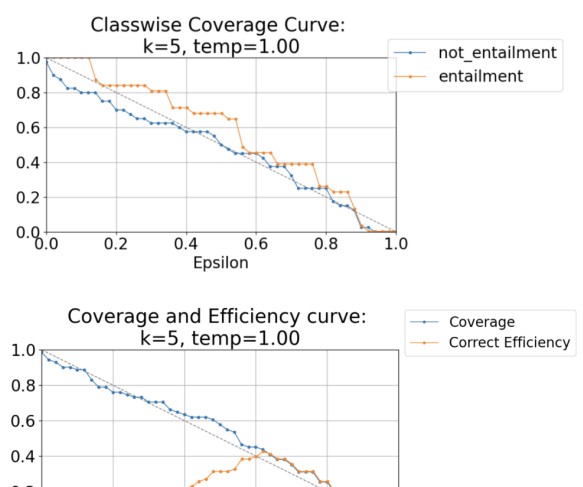

Figure 39: BERT-small WNLI classwise-false

Figure 40: BERT-small WNLI classwise-true

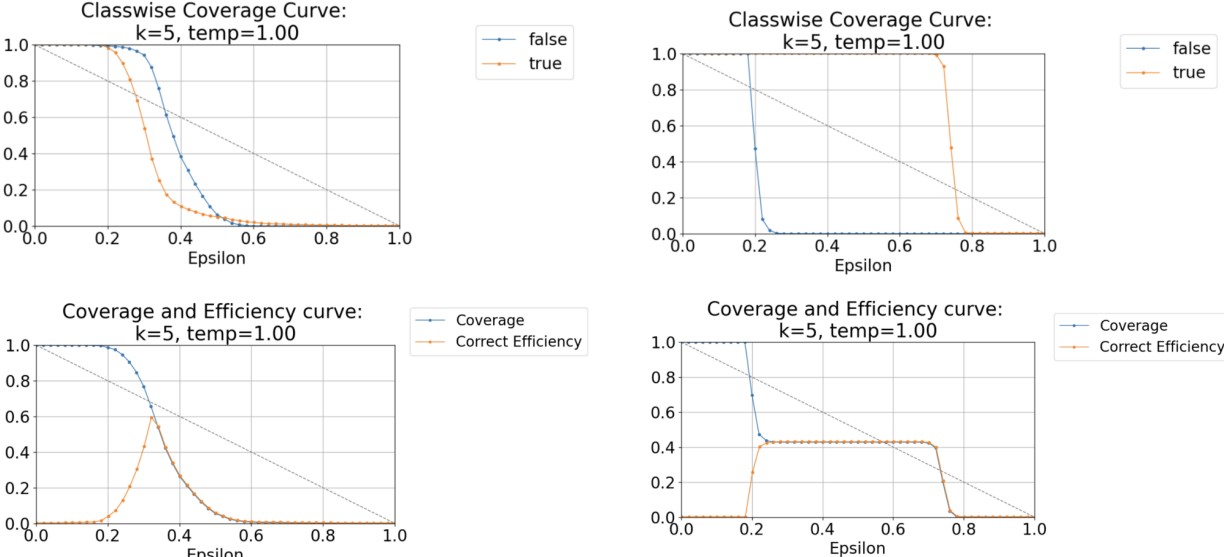

Figure 41: BERT-tiny MultiRC classwise-false          Figure 42: BERT-tiny MultiRC classwise-true

