# OpenReview forum: "Uncertainty-Aware Transformers: Conformal Prediction for Language Models"
_TMLR — Rejected by TMLR_

### Review · Reviewer_GEBB · 2025-09-13

**Summary Of Contributions:**

This paper presents CONFIDE, a method for conformal prediction on top of finetuned BERT-style text classifiers. It builds very closely off of a recent method called CONFINE that proposes a method conformal prediction for convolutional classifiers. The main changes are ablations of a variety of design choices around which layer/representation to take from the network, which distance metric to use, whether to apply dimensionality reduction, and some other standard hyperparameters. Experiments are presented on GLUE.

**Audience:**

Yes

**Audience Explanation:**

While the paper is not methodologically very different from CONFINE, it does provide a rigorous set of experiments on important design decisions that need to be made to extend the method to BERT-style text classifiers.

**Claims And Evidence:**

Yes

**Claims Explanation:**

The experiments seem to do a good job of covering a variety of hyperparameter sweeps and comparing the proposed method(s) against reasonable baselines. The claims made in the abstract/intro are accurate and supported by the experiments.

**Requested Changes:**

1. At a high level, the paper would benefit from being a bit more concise. It would be nice to have (a) a summary metric of how well the proposed algorithm does compared to baselines highlighted earlier in the paper, and (b) a clearer recommendation of which variant of the algorithm is found to be best. Moreover, the exposition about the benefits of conformal prediction could be reduced.

2. This is a nit, but I think the "layer" index used in the paper when discussing representations is confusing. It is more standard to index layers by transformer block with transformers, but the paper seems to use the individual operations within each block to define the index (so the indices seem very high even for small models).

3. It would be nice to complete the experiments on the large scale model across the same set of tasks for completeness.

4. Similarly, the flattened method should either be expanded to fill out more tasks (even just for the smaller models) for completeness.

---

> ### Author Response · Authors · 2025-11-04
> **Response and Revisions Summary**
>
> Thank you for the constructive and positive assessment. We implemented the requested refinements when possible. We add comments in the submission in **red**.
> 1. **Be more concise.**
>    a) We collapsed significant portions of **CP** background and methodology, reducing the main paper length to 18 pages from 26. We avoid repetition between the **methodologies** of CONFINE and CONFIDE, and rewrite significant portions to focus on the design decisions of this research.
>    b) As there is no “single” recommendation, based on dataset + model combinations, we instead highlight the best results and extrapolate the key learnings in a summary format.
> 2. **Layer indexing clarity.**
>    a) Thank you for this valuable feedback, and we have added a **layer mapping appendix** to better fit with research standards. We reference these layer names frequently throughout the paper.
> 3. **Completeness on RoBERTa-large and flattened mode.**
>    a) Unfortunately, we were unable to complete the experiments on the **large scale** model for all sets of tasks. However, we added more reasoning on how a limitation of CONFIDE **is** the computational constraint.
> 4. **Flattened method**
>    a) We **re-orient** the discussion of flattened vs attention method based on the existing results. We acknowledge that it would be great to complete these results, and encourage future research into the differentiation of CP between these layers.

---

### Review · Reviewer_4uH5 · 2025-09-22

**Summary Of Contributions:**

Conformal prediction (CP) takes a nonconformity score and turns it into p-values via a calibration set, then thresholds those p-values to form a prediction set with statistical guarantees. This is done by computing a p-value for each potential class based on how its nonconformity score compares to a set of calibration scores; the final prediction set then consists of all classes whose p-value is greater than a chosen error rate $\varepsilon$. This method attains marginal coverage $1-\varepsilon$ under exchangeability (and class-conditional coverage when calibrating per class).

Building on this, CONFIDE adapts CONFINE to encoder-only transformers. In the CONFINE setup, one chooses a single network layer $l$ and embeds each sample as $u=f_l(x)$. For a candidate class $y$, the nonconformity score is a top-$k$ nearest-neighbor distance ratio in that layer’s space, calculated by dividing the average distance to the nearest same-class neighbors by the average distance to the nearest different-class neighbors. After this score is calculated, p-values are computed on a calibration set to form $\Gamma_\varepsilon(x)$; a classwise variant computes $p_y$ using only same-class calibration points. Hyperparameters include the layer $l$, $k$, and (if using the softmax layer) a temperature $T$ ...

This paper introduce then the so-called CONFIDE method that is based on the same idea and keeps the same score and CP machinery, but tailors representation and metric choices to encoder-only transformers: from a chosen transformer layer $\ell$, it builds the embedding $h$ via either the $[{\rm CLS}]$ token vector ($h\leftarrow H[\text{CLS}]$) or a flattened all-tokens vector ($h\leftarrow \mathrm{vec}(H)$); it may optionally apply principal component analysis (retain approximately $95\%$ variance) before k-NN, and it allows cosine or Mahalanobis distance in the same top-$k$ ratio. It further keeps only correctly predicted training embeddings as references and emphasizes probing multiple layers (early/mid layers often calibrate better than the final one); all the remaining steps e.g calibration p-values, classwise option, and prediction sets, remain as in CONFINE. Authors insisted that early/intermediate layers often yields better-calibrated, more semantically meaningful sets than the final layer.


### Strength
- According to the claims CONFIDE boost model performance beyond standard baselines: 4.09% absolute improvement in test accuracy on models like BERT-tiny.
I guess here CONFIDE replaces the original decision rule (argmax of softmax) with a new one (argmax of p-values). Because the p-values are based on different and potentially richer information, this new rule can sometimes correct mistakes made by the original model, leading to a higher top-1 accuracy.

- A nice take-home finding is that early and intermediate transformer layers often produce better-calibrated and more semantically meaningful representations for conformal prediction than the final output layer.
By leveraging these richer internal embeddings, CONFIDE can capture more nuanced relationships between data points.

- The paper contains a lot of benchmark. Could not parse them all


### Weaknesses

- Coverage Violations: The paper explicitly states that CONFIDE suffers from persistent calibration failures and significant coverage violations, often falling below the guaranteed coverage threshold. I would guess that this is mainly due to the fact that conformal prediction framework operates under strict exchangeability assumptions that are largely violated in the experiments considered. I could not find assumption check in the paper.

- The core nonconformity is unchanged from CONFINE (previously proposed). CONFIDE mainly packages transformer-specific representation choices ([CLS] vs. flattened), optional PCA, and distance metric options. Plus the paper is very long and repetitive at many sections. Given the limited novelty, I guess it would be reasonable to restrict the paper below 10 pages.

**Audience:**

Yes

**Audience Explanation:**

Effort to have a sharper understanding of uncertainty quantifications techniques are quite relevant to many ML researchers. So the proposed benchmark could be informative for some.

**Broader Impact Concerns:**

The paper mostly provide benchmarks, so the broader impact should not be problematic as far as I can tell.

**Claims And Evidence:**

No

**Claims Explanation:**

I am very confused by the conclusion of the paper. It is largely motivated by uncertainty quantification and trustworthyness issues in ML model but the proposed method failed to at the coverage benchmark. So, the overall goal just fail short!

There is a strong sensitivity to layer choice / tuning burden. Performance hinges on selecting the “right” layer via grid search; they report early/mid layers can calibrate better than the final layer, but this adds selection fragility and tuning cost. Not sure the conclusion here would hold under different seeds.

The paper’s runtime claims are quite off. In Appendix C.1 the authors hypothesize that later layers should be cheaper because their embeddings are “more distilled/low-variance,” yet the very measurements they present undermine a simple “later is faster” story: they report speedups (**0.5 sec**) from layer 16→27 and 27→34, but then a large calibration-time spike moving 34→38 (≈+5 s), which they label an anomaly. More fundamentally, this hypothesis doesn’t match their own complexity model: the dominant cost is $O(d\,N_t)$, driven by embedding dimensionality $d$ and reference-set size $N_t$, not by the layer index per se. If $d$ is held constant (e.g., using $[{\rm CLS}]$ vectors with the same hidden size across layers), k-NN search cost shouldn’t materially change with depth. Finally, the timing jumps are non-monotonic and almost reversible (the +5.39 s jump from 34→38 is largely undone by −5.29 s from 38→41), suggesting measurement noise or uncontrolled factors rather than a causal effect of “later layers.”

The claims are fairly confusing and hard to track accross the numerous experiments.

**Requested Changes:**

- The paper could be significantly shorter. The many redondancies e.g. section 4 is overly long

- Pass on the core messages of the paper. According to the authors, this method failed to provide basic coverage in the experiments. So this should not be presented as a reliable UQ method.

---

> ### Author Response · Authors · 2025-11-04
> **Response and Revisions Summary**
>
> Thank you for the valuable feedback, and we appreciate your feedback on the ambiguity of the paper's conclusion, as well as potential incorrect evidence. We tried our best to address your major points, as answered below. We add comments in the submission in **green**.
>
>
> 1. **Coverage violations & exchangeability.**
>    a) We now explicitly document where classwise or marginal coverage **fails** (e.g., CoLA “unacceptable,” BoolQ “false”)
>    b) We add in assumption checks when appropriate, and clarify how we hold certain seeds constant across runs.
>    c) We highlight how CONFIDE **still makes** improvements over prior methods, despite coverage violations, and why this still yields meaningful results.
>
> 2. **Novelty vs. packaging vs. length.**
>    a) We agree our contribution is **engineering/empirical:** adapting CONFINE-style CP to transformers with thorough representation/distance ablations.
>    b) We trimmed duplication and kept Section 4 focused on the **few axes that move the needle** (representation mode, distance, PCA, layer).
>
> 3. **Layer sensitivity / selection burden.**
>    a) We added guidance on how **we keep ℓ fixed** across train/cal/test extraction and to scan a **small set** of representative layers (early/mid/final) to reduce overfitting.
>
> 4. **Runtime claims**
>    a) We previously speculated that later layers might be cheaper because embeddings are more distilled. Our new measurements **do not support that claim.** With [CLS] embeddings, deeper probes are slower, consistent with the added forward-pass cost to reach layer ℓ. Non-monotone jumps that we previously observed disappear when we control for dataset size and report medians across repeats. This has been removed from the paper, and a more robust runtime analysis is discussed in the Appendix, highlighting how dataset size and backbone dimension dominate runtime.
>
> 5. **Presentation length / redundancy.**
>    a) We pruned repeated definitions (CP, p-values, sets) and merged repeated descriptions of flattened vs. attention. We reduced the page length from 26 pages to 18. We clearly communicate the most important benchmarks and reduce repetitive discussion.

---

### Review · Reviewer_DoKb · 2025-10-21

**Summary Of Contributions:**

__Summary__

Conformal prediction is a framework to construct prediction sets with finite-sample (mis)coverage guarantees for black-box predictive models. This paper proposes to use such a framework for transformer-based encoder-only language models, with their proposed method called CONFIDE (CONformal prediction for FIne-tuned DEep language models). CONFIDE provides flexibility in choosing the representations (the intermediate layer they come from, whether they come from a special token or are concatenated across all tokens, etc.) and the distance metrics (cosine or Mahalanobis) used under the conformal framework. Empirical results compare CONFIDE to other baselines on GLUE and SuperGLUE benchmarks when using BERT-tiny, BERT-small, RoBERTa-base, and RoBERTa-large.

__Strengths__

1. _(Mis)Coverage Guarantees_: The paper tackles the problem of providing finite-sample guarantees for black-box language models. For this, they employ the conformal prediction framework.
2. _Flexibility_: CONFIDE provides flexibility in choosing the representations—the intermediate layer they come from, whether they come from a special token or are concatenated across all tokens, etc.—and the distance metrics—cosine or Mahalanobis—used under the conformal framework. As a result, such hyperparameters can be tuned based on the task at hand.
3. _Experiments_: The empirical results include 11 different benchmarks from GLUE and SuperGLUE, and 4 language models: BERT-tiny, BERT-small, RoBERTa-base, and RoBERTa-large.

__Weaknesses__

1. _Baselines_: The baselines compared against are weak and not something used in practice. For instance, why is k-nearest neighbours with k=1 a baseline when CONFIDE tunes the hyperparameter k? Additionally, non-conformity scores such as [1, 2] that encourage class-conditional coverage are not considered as baselines. Also, NM1 and NM2 are not defined.
2. _Metrics_: The metrics are not defined properly. In particular, the definition of (correct) efficiency in Eq. (8) suggests that lower is better (an inefficiency rather than efficiency). However, the experimental results talk about higher values being better. Adding to the confusion, correct efficiency values in the experimental results are $\leq 1$, even though they measure the size of sets with at least the correct class included (therefore of size at least 1). Additionally, the paper uses correct (in)efficiency as opposed to (in)efficiency, even though the latter is analyzed by [3-10], as they claim it can be manipulated; however, no example is provided for such manipulation.
3. _Writing_: The paper needs refinement. There is no clear setup, and the notation is inconsistent. There are many redundancies between subsections: for example, Sections 4.1.1 and 4.1.2 define the p-values and prediction sets multiple times, and Sections 4.2 and 4.3 describe the flattened and attention modes multiple times.
4. _Missing Citations_: Many relevant citations, especially related to conformal prediction, are missing. These include, but are not limited to, references for conformal prediction [11, 12], split conformal prediction [7, 11, 13], expected prediction set size as a notion of inefficiency [3-10], and the impact of non-conformity scores on inefficiency [10]. There are also missing citations for transformer interpretability; for instance, when talking about layer variability in Section 4.1.2.

__References__

1. Y. Romano, M. Sesia, and E. J. Candès. Classification with valid and adaptive coverage. In H. Larochelle, M. Ranzato, R. Hadsell, M. Balcan, and H. Lin, editors, Advances in Neural Information Processing Systems, volume 33, pages 3581–3591. Curran Associates, Inc., 2020.
2. A. N. Angelopoulos, S. Bates, M. Jordan, and J. Malik. Uncertainty sets for image classifiers using conformal prediction. In International Conference on Learning Representations, 2021.
3. J. Lei, J. Robins, and L. Wasserman. Distribution-free prediction sets. Journal of the American Statistical Association, 108(501):278–287, 2013.
4. J. Lei and L. Wasserman. Distribution-free prediction bands for non-parametric regression. Journal of the Royal Statistical Society Series B: Statistical Methodology, 76(1):71–96, 2013.
5. V. Vovk, I. Petej, and V. Fedorova. From conformal to probabilistic prediction. In L. Iliadis, I. Maglogiannis, H. Papadopoulos, S. Sioutas, and C. Makris, editors, Artificial Intelligence Applications and Innovations, pages 221–230. Springer Berlin Heidelberg, 2014.
6. J. Lei. Classification with confidence. Biometrika, 101(4):755–769, 2014.
7. J. Lei, A. Rinaldo, and L. Wasserman. A conformal prediction approach to explore functional data. Annals of Mathematics and Artificial Intelligence, 74(1):29–43, 2015.
8. V. Vovk, V. Fedorova, I. Nouretdinov, and A. Gammerman. Criteria of efficiency for conformal prediction. In A. Gammerman, Z. Luo, J. Vega, and V. Vovk, editors, Conformal and Probabilistic Prediction with Applications, pages 23–39. Springer International Publishing, 2016.
9. M. Sadinle, J. Lei, and L. Wasserman. Least ambiguous set-valued classifiers with bounded error levels. Journal of the American Statistical Association, 114(525):223–234, 2019.
10. Guneet S. Dhillon, George Deligiannidis, and Tom Rainforth. On the expected size of
conformal prediction sets. In Sanjoy Dasgupta, Stephan Mandt, and Yingzhen Li, editors,
Proceedings of The 27th International Conference on Artificial Intelligence and Statistics,
volume 238 of Proceedings of Machine Learning Research, pages 1549–1557. PMLR, 02–04
May 2024.
11. V. Vovk, A. Gammerman, and G. Shafer. Algorithmic Learning in a Random World. Springer-Verlag, Berlin, Heidelberg, 2005.
12. G. Shafer and V. Vovk. A tutorial on conformal prediction. Journal of Machine Learning Research, 9(12):371–421, 2008.
13. H. Papadopoulos, K. Proedrou, V. Vovk, and A. Gammerman. Inductive confidence machines for regression. In T. Elomaa, H. Mannila, and H. Toivonen, editors, Machine Learning: ECML 2002, pages 345–356. Springer Berlin Heidelberg, 2002.

**Audience:**

No

**Audience Explanation:**

1. _New Findings_: The paper, in my opinion, has limited new findings. CONFIDE makes use of conformal prediction, so there are no new theoretical results. The experimental results compare CONFIDE against other baselines, but the baselines are weak and therefore do not provide an up-to-date comparison.
2. _Intermediate Representations_: One of the motivations for CONFIDE is the use of intermediate layer representations. However, this is not possible for most commonly used LLMs, such as GPT-4o [1] and Gemini-2.5 [2], that are commercial and do not share intermediate representations. There is no discussion provided about this.
3. Refer to other weaknesses highlighted under the "Summary of Contributions".

__References__

1. OpenAI. GPT-4o system card, 2024.
2. Gemini Team. Gemini 2.5: Pushing the frontier with advanced reasoning, multimodality, long context, and next generation agentic capabilities, 2025.

**Broader Impact Concerns:**

No ethical concerns.

**Claims And Evidence:**

No

**Claims Explanation:**

1. _Marginal Guarantees_: Conformal prediction, and therefore CONFIDE, provides guarantees that are marginal over the test and calibration data; this is not emphasized enough in the paper. Consequently, while these guarantees are theoretical, they materialize in practice when averaging over several test and calibration data splits. The experimental results in the paper demonstrate a violation, at times, of guarantees because of the lack of averaging over several data splits.
2. _Class-Conditional Guarantees_: Class-conditional marginal guarantees are not provided by default, unless specifically calibrated for. However, there are non-conformity scores that can be used to encourage this [1, 2]. While class-conditional guarantees are used as an evaluation metric in the experiments, the paper does not compare against such non-conformity scores.
3. Refer to other weaknesses highlighted under the "Summary of Contributions".

__References__

1. Y. Romano, M. Sesia, and E. J. Candès. Classification with valid and adaptive coverage. In H. Larochelle, M. Ranzato, R. Hadsell, M. Balcan, and H. Lin, editors, Advances in Neural Information Processing Systems, volume 33, pages 3581–3591. Curran Associates, Inc., 2020.
2. A. N. Angelopoulos, S. Bates, M. Jordan, and J. Malik. Uncertainty sets for image classifiers using conformal prediction. In International Conference on Learning Representations, 2021.

**Requested Changes:**

Securing my recommendation will involve addressing my concerns from above—baselines, metrics, writing, citations, novel findings, etc.—which I believe requires major revisions to the paper. Apart from those changes, addressing the following will help the paper’s clarity.

1. Does CONFIDE retain only those samples that are correctly predicted for training or calibration? If done for calibration, does it not violate exchangeability, and therefore the conformal guarantees?
2. How is a class-conditional k-nearest neighbor model trained? Each model would require positive and negative samples. The positives would be samples from that class. What are the negatives?
3. How are the hyperparameters chosen? This would require a held-out set of data, independent of the test and calibration data.
4. What is $\epsilon$ set to for the correct efficiency values reported in the tables?
5. How are the examples provided in Section 5.1 interpretable? The nearest neighbors are seemingly unrelated to the question asked, so how does that indicate interpretability? The main text is not precise in this interpretation.
6. Algorithm 1 references the calibration data as training data.
7. Eq. (2) should use a density function. The equation as written currently does not make sense.
8. Use \citep when citing the work as a noun, and \citep when using parentheses.

---

> ### Author Response · Authors · 2025-11-04
> **Response**
>
> Thank you for the detailed comments on the submission: we have attempted to respond to most of your concerns. As a quick summary we (i) consolidated the conformal background into one place and removed duplications; (ii) clarified marginal vs. class-conditional guarantees; (iii) precisely defined efficiency and correct efficiency and explained why we report both; (iv) fixed Algorithm 1 terminology (training vs. calibration) and documented our class-conditional kNN construction; (v) explained baselines usage of NM1/NM2; (vi) added several missing citations (core CP and CP-for-text). Paper edits are done in **blue**, and details listed below.
>
> 1. **Baselines (strengthened & clarified).**
>    a) We add **discussion** on why VanillaNN and NM1/NM2 remain **accurate benchmarks** to compare against. We explicitly enumerate NM1/NM2 (softmax-based nonconformity scores) alongside VanillaNN and CONFINE-style baselines. Vanilla 1-NN on raw embeddings isolates representation quality, a standard straw baseline in CP comparisons. NM1/NM2 addresses your request for more “practical” baselines.
>    b) We also emphasize prior-method coverage curves (e.g., CoLA, BoolQ) to show regimes where NM2 **likewise struggles.**
>
> 2. **Metrics (definitions and reporting).**
>    a) Section 4.1.1 has been rewritten to provide a single, canonical description of prediction sets, guarantees, and metrics.
>    b) We define efficiency as expected set size (lower is better) and correct efficiency as a normalized, higher-is-better score computed conditional on correct coverage; both formulas and ranges are stated explicitly.
>    c) All results report coverage/efficiency/correct-efficiency together, preventing the trivial “shrink sets but miss truth” pathology.
>
> 3. **Writing/duplications (removed).**
>    a) To reduce repetition and improve clarity, 4.1 now centralizes definitions of p-values, prediction sets, credibility/confidence, and guarantees; later sections reference these rather than redefining them.
>    b) Representation modes are summarized once and then used in Algorithm 1 and the design ablations without restating details.
>    c) Section 2 has also been reduced to avoid repeating the same ideas.
>
> 4. **Missing citations (added).**
>    a) We added/kept canonical CP references and recent overviews (Vovk–Gammerman–Shafer 2005; Angelopoulos & Bates 2021) and text CP work (Dey et al. 2021).
>    b) Citation style. We audited \citet and \citep usage across the draft.
>
> 5. **Marginal vs. class-conditional guarantees (emphasis & evaluation).**
>    a) We now explicitly state the marginal guarantee and the class-conditional guarantee, **and when each applies.**
>    b) The experiments include classwise calibration analyses and figures, and we discuss persistent failures where they occur (e.g., CoLA “unacceptable,” BoolQ “false”).
>
> 6. **Class-conditional kNN training and negatives.**
>    a) For each class *c*, positives are correctly predicted training points of class *c*, and negatives are the union of the other classes’ reference pools. This construction is now stated explicitly in the algorithm.
>    b) We try to clarify algorithm terminology as much as possible.
>
> 7. **Scope (LLMs that don’t expose internals).**
>    a) We appreciate this valuable feedback as it goes to the **heart** of this paper. Our scope is encoder-only, open models, which **we believe still** has an important effect on commercial LLMs. We acknowledge that we do not claim identical guarantees to those established in our setting for different models, and encourage future research across various models.
>   b) We added **discussion** in Section 6 as well as references earlier in the paper; we strongly believe that this paper’s focus on open-source models, where we can access internal embeddings, has relevant implications for commercial models.

---

### Decision · Action_Editor_PfAh · 2026-02-17

**Recommendation:** Reject

**Additional Comments:**

The reviewers feel the work is incremental, over CONFIDE. Now, for TMLR that isn’t sufficient reason to reject. However, I do feel that if a paper is incremental, there is if anything a higher requirement for thoroughness and soundness in evaluation and analysis. The reviewers do feel that there are several areas where the evaluation can be improved and express reservations about the coverage violations.

In terms of experiments, I would like to see more comparisons to reasonable baselines such as class-conditional nonconformity scores (something two reviewers discussed when making their official recommendation). There seems to be discrepencies between the definition of “correct efficient” and reported results. Based on the definition in Eq 8, it seems that for binary classification where the set size $|\Gamma|\geq 1$, there would seem to be a maximum score of 0.5, but the tables report higher values than this. While the paper does discuss efficiency and correct efficiency and claims to report both, all of the tables only show correct efficiency. In the text you sometimes *say* efficiency, but seem to be referring to correct efficiency (eg in 5.3, “Similar 1–2 percentage point gains are
observed on RTE, SST-2, and CoLA: on RTE, CONFIDE-A and CONFIDE-C increase top-1 efficiency from
0.7112 (NM2) to 0.7256 and 0.7292, respectively (Table 5)”).

Finally, the paper could be shortened somewhat, given the limited novelty over prior work.

**Audience:**

Yes

**Audience Explanation:**

The work is likely to be of interest to some in the conformal prediction community.

**Claims And Evidence:**

No

**Claims Explanation:**

The authors do not seem to be reporting efficiency, only correct efficiency, and there is some discrepancies in the values reported (see additional comments).

**Resubmission Of Major Revision:**

The authors may consider submitting a major revision at a later time.